



# Estimating Emissions of Methane Consistent with Atmospheric Measurements of Methane and $\delta^{13}$C of Methane

Sourish Basu[1,2], Xin Lan[3,4], Edward Dlugokencky[4], Sylvia Michel[5], Stefan Schwietzke[6], John B. Miller[4], Lori Bruhwiler[4], Youmi Oh[4], Pieter P. Tans[4], Francesco Apadula[7], Luciana V. Gatti[8], Armin Jordan[9], Jaroslaw Necki[10], Motoki Sasakawa[11], Shinji Morimoto[12], Tatiana Di Iorio[13], Haeyoung Lee[14], Jgor Arduini[15], and Giovanni Manca[16]

[1]Global Modeling and Assimilation Office, NASA Goddard Space Flight Center, Greenbelt MD
[2]Earth System Science Interdisciplinary Center, University of Maryland, College Park MD
[3]Cooperative Institute for Research in Environmental Science, University of Colorado, Boulder CO
[4]Global Monitoring Laboratory, National Oceanic and Atmospheric Administration, Boulder CO
[5]Institute for Arctic and Alpine Research, University of Colorado, Boulder CO
[6]Environmental Defense Fund, Berlin, Germany
[7]Ricerca sul Sistema Energetico (RSE S.p.A.), Milano, Italy
[8]Instituto Nacional de Pesquisas Espaciais, São José dos Campos, São Paulo, Brazil
[9]Max Planck Institute for Biogeochemistry, Jena, Germany
[10]AGH University of Science and Technology, Krakow, Poland
[11]National Institute for Environmental Studies, Tsukuba-shi, Ibaraki, Japan
[12]Center for Atmospheric and Oceanic Studies, Tohoku University, Sendai, Japan
[13]Italian National Agency for New Technologies, Energy, and Sustainable Economic Devlopment (ENEA), Rome, Italy
[14]National Institute of Meteorological Sciences, Seogwipo-si, Jeju-do, Korea
[15]Università degli Studi di Urbino, Urbino, Italy
[16]European Commission, Joint Research Center, Ispra, Italy

**Correspondence:** Sourish Basu (sourish@umd.edu)

**Abstract.** We have constructed an atmospheric inversion framework based on TM5 4DVAR to jointly assimilate measurements of methane and $\delta^{13}$C of methane in order to estimate source-specific methane emissions. Here we present global emission estimates from this framework for the period 1999–2016. We assimilate a newly constructed, multi-agency database of $CH_4$ and $\delta^{13}CH_4$ measurements. We find that traditional $CH_4$-only atmospheric inversions are unlikely to estimate emissions consistent

5   with atmospheric $\delta^{13}CH_4$ data, and assimilating $\delta^{13}CH_4$ data is necessary to deriving emissions consistent with both measurements. Our framework attributes *ca.* 85% of the post-2007 growth in atmospheric methane to microbial sources, with about half of that coming from the Tropics between 23.5 °N and 23.5 °S. This contradicts the attribution of the recent growth in the methane budget of the Global Carbon Project (GCP). We find that the GCP attribution is only consistent with our top-down estimate in the absence of $\delta^{13}CH_4$ data. We find that at global and continental scales, $\delta^{13}CH_4$ data can separate microbial from fossil methane emissions much better than $CH_4$ data alone can, and at smaller scales this ability is limited by the current

10   $\delta^{13}CH_4$ measurement coverage. Finally, we find that the largest uncertainty in using $\delta^{13}CH_4$ data to separate different methane source types comes from our knowledge of atmospheric chemistry, specifically the distribution of tropospheric chlorine and the isotopic discrimination of the methane sink.





# 1 Introduction

Current atmospheric levels of methane (CH$_4$) are about 2.5 times pre-industrial levels, primarily due to anthropogenic emissions (Dlugokencky et al., 2011). The main sources of CH$_4$ to the atmosphere today are known, which are periodically summarized by the Global Carbon Project (GCP, Saunois et al., 2020). In brief, they include anthropogenic sources from agriculture (ruminants, manure, and rice), waste management (landfills and waste treatment), fossil fuel production and use (coal, oil, and natural gas), and biomass burning (including biofuels). The remainder is from natural processes, predominantly tropical and high northern latitude wetlands, with smaller contributions from termites, wild animals and geologic seeps. In the latest GCP report, however, there remains a large disparity of $\sim$160 Tg yr$^{-1}$ between the bottom-up budget constructed from inventories and the top-down budget derived from atmospheric CH$_4$ measurements (Saunois et al., 2020), signifying gaps in our understanding of the CH$_4$ budget.

As shown in Figure 1, CH$_4$ levels have been rising rapidly since 2007 after a period of relatively slow growth in 1999–2006 (Dlugokencky et al., 2011; Saunois et al., 2020). The mechanisms behind the relative stability of 1999–2006 and growth thereafter, however, are not yet fully understood. Possible mechanisms suggested in the literature include an approach to a steady state in the early 2000s (Dlugokencky et al., 2003) followed by an increase in either agricultural (Schaefer et al., 2016) or fossil (Worden et al., 2017) emissions, or both (Saunois et al., 2020; Jackson et al., 2020); increase in global (Schwietzke et al., 2016) or tropical (Nisbet et al., 2016, 2019) microbial emissions; a decrease in methane uptake by upland soils (Ni and Groffman, 2018); and decadal changes in the atmospheric sinks of methane (Rigby et al., 2017; Turner et al., 2017). It is difficult to choose between these competing explanations based on atmospheric CH$_4$ measurements alone. However, measurements of the $^{13}$C:$^{12}$C ratio of CH$_4$, denoted $\delta^{13}$C$-$CH$_4$ or $\delta^{13}$CH$_4$ in short, provide some additional information to distinguish between these hypotheses (Lan et al., 2021).

Different CH$_4$ sources have distinct $\delta^{13}$CH$_4$ signatures over large spatial scales, and different sinks consume $^{12}$CH$_4$ and $^{13}$CH$_4$ at slightly different rates, imposing different signals on atmospheric $\delta^{13}$CH$_4$ (Miller, 2004). Therefore, atmospheric $\delta^{13}$CH$_4$ measurements can help constrain and refine the CH$_4$ budget. In an earlier publication, we described the simulation of atmospheric CH$_4$ and $\delta^{13}$CH$_4$ using the model TM5 (Krol et al., 2005) and its use for evaluating competing hypotheses about renewed CH$_4$ growth since 2007 (Lan et al., 2021). In this work, we construct and apply a variational inversion framework based on TM5 to assimilate CH$_4$ and $\delta^{13}$CH$_4$ measurements and estimate space- and time-varying emissions of CH$_4$ disaggregated by source type. With this framework, we perform atmospheric inversions from 1997 to 2016 to infer large scale methane emissions from different sources, assess the added value of $\delta^{13}$CH$_4$ measurements compared to traditional CH$_4$-only inversions, and investigate the possible factors behind the post-2007 growth in atmospheric CH$_4$.

Several previous studies have used the information provided by $\delta^{13}$CH$_4$ measurements to infer mechanisms behind the behavior of atmospheric methane over the past two decades. However, many of these studies approximated the global atmosphere as a small number of connected boxes, with homogeneous emissions and chemistry in each box (e.g., Schwietzke et al., 2016; Schaefer et al., 2016; Nisbet et al., 2016, 2019; Worden et al., 2017). They were therefore susceptible to biases inherent in box models (Naus et al., 2019) and were unable to use the information contained in spatial gradients of atmospheric CH$_4$





and $\delta^{13}CH_4$. Moreover, by construction, box models have to simplify the complexity of $\delta^{13}CH_4$ source signatures, transport variability and loss processes, and cannot extract information from spatial gradients in atmospheric measurements. Some stud-

50 ies have used 3D atmospheric circulation models to estimate $CH_4$ emissions consistent with observed $\delta^{13}CH_4$ (e.g., Bousquet et al., 2006; Rice et al., 2016). However, they have generally used globally uniform $\delta^{13}CH_4$ source signatures, when in reality signatures of some of the most important sources such as wetlands and fossil fuels have strong latitudinal gradients and spatial variations. In this study, we confront our best estimate of spatio-temporally varying methane emissions and source signatures with a newly constructed multi-laboratory dataset of atmospheric $CH_4$ and $\delta^{13}CH_4$ measurements in the TM5 4DVAR frame-

55 work. Our technique is analogous to a recently submitted manuscript by Thanwerdas et al. (2021), and in § 4.1 we discuss the similarities and differences between our methods.

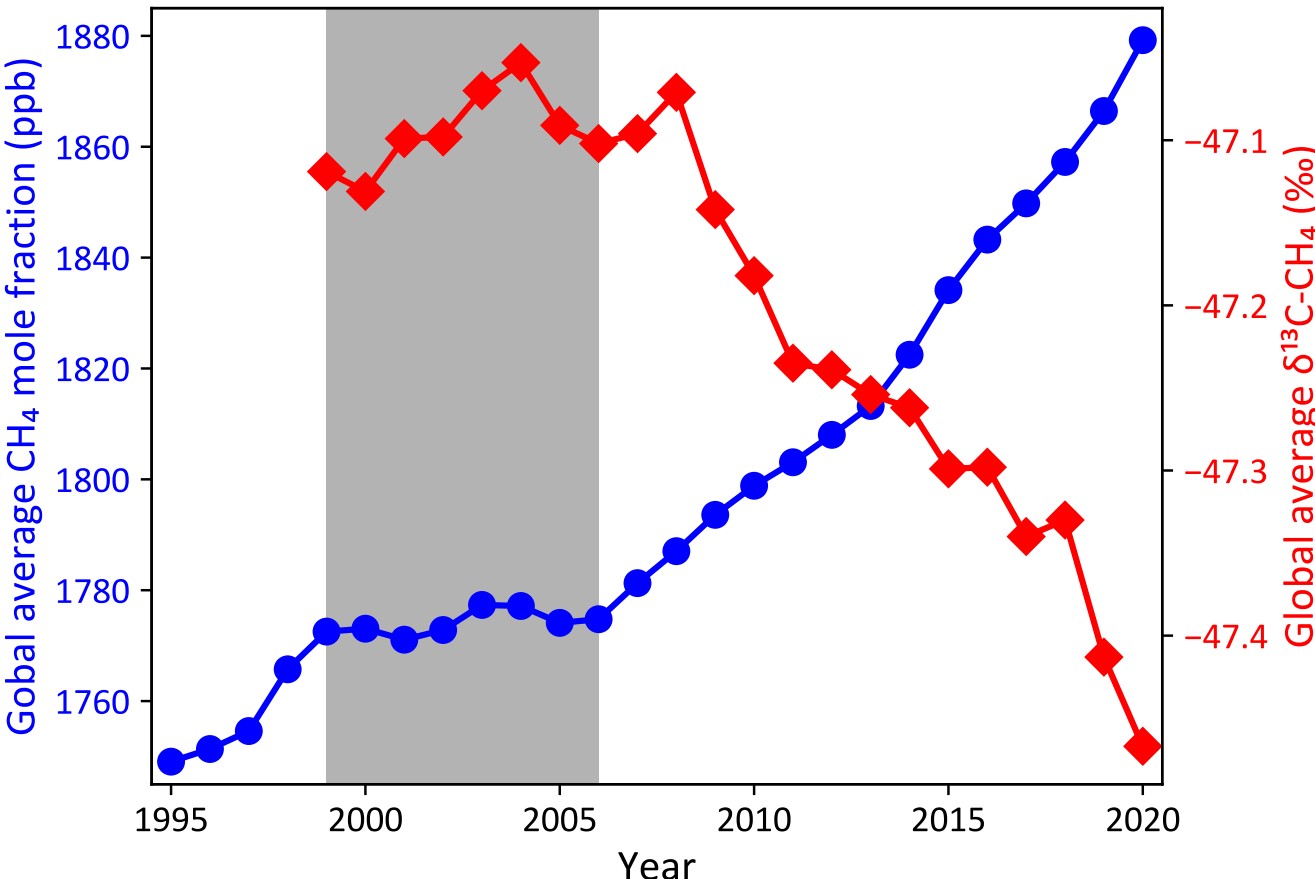

**Figure 1.** Global average $CH_4$ (blue circles, left axis) and $\delta^{13}CH_4$ (red diamonds, right axis) from NOAA marine boundary layer (MBL) and other background sampling sites. The gray box denotes the period from 1999 to 2006 when the atmospheric $CH_4$ burden was relatively stable, in contrast to the periods of growth before and after. Regular $\delta^{13}CH_4$ measurements started at NOAA background sites in 1998, which is the first year with an estimate of the global $\delta^{13}CH_4$. The selection of marine boundary layer sites and the construction of global averages is described in detail by Masarie and Tans (1995).



## 2 Method

### 2.1 Formulation of the mass balance equations

The atmospheric mass balance of $^{12}\mathrm{CH}_4$ can be written as

$$\frac{d}{dt}\,^{12}\mathrm{C} = \sum_s \,^{12}\mathrm{F}_s - \sum_l \,^{12}\mathrm{k}_l \times \,^{12}\mathrm{C} \times [l]$$

$$= \sum_s \,^{12}\mathrm{F}_s - \,^{12}\mathrm{C} \sum_l L_l \tag{1}$$

where $s$ denotes three isotopically distinct source categories, namely pyrogenic, fossil and microbial $\mathrm{CH}_4$; and $l$ denotes species contributing to chemical loss, namely Cl, OH and $\mathrm{O}(^1\mathrm{D})$. While the upland soil sink of methane is included in the sources in conventional methane modeling, for reasons described in § 2.4 we have modeled it as a fourth loss mechanism. The combination $^{12}\mathrm{k}_l \times [l]$ can be denoted as an inverse lifetime $L_l$ due to species $l$. For $^{13}\mathrm{CH}_4$, we can write a corresponding equation,

$$\frac{d}{dt}\,^{13}\mathrm{C} = \sum_s \,^{13}\mathrm{F}_s - \,^{13}\mathrm{C} \sum_l \alpha_l L_l \tag{2}$$

where $\alpha_l = \,^{13}\mathrm{k}_l / \,^{12}\mathrm{k}_l$. Using the definition of $\delta$, we can substitute $^{13}\mathrm{C} = \,^{12}\mathrm{C} r_{\mathrm{std}}(\delta_a + 1)$ and $^{13}\mathrm{F} = \,^{12}\mathrm{F} r_{\mathrm{std}}(\delta_s + 1)$ and get

$$^{12}\mathrm{C}\frac{d}{dt}\delta_a = \sum_s (\delta_s - \delta_a)\,^{12}\mathrm{F}_s - \,^{12}\mathrm{C}(\delta_a + 1)\sum_l \epsilon_l L_l \tag{3}$$

where $\epsilon_l = \alpha_l - 1$ and $r_{\mathrm{std}} = 0.0112372$ is a pre-defined standard ratio[1] (Craig, 1957). While equations (1) and (3) are mathematically complete descriptions of the $^{12}\mathrm{CH}_4$ and $^{13}\mathrm{CH}_4$ budgets, they are not the most convenient form for constructing a dual tracer $\mathrm{CH}_4$ and $\delta^{13}\mathrm{CH}_4$ inversion. This is because it is total $\mathrm{CH}_4$ that is measured and not the two isotopologues separately. We therefore construct an alternate formulation in terms of $\delta' = (^{13}\mathrm{CH}_4/\mathrm{CH}_4)/r_{\mathrm{std}} - 1$, which can be related to the more traditional $\delta = (^{13}\mathrm{CH}_4/^{12}\mathrm{CH}_4)/r_{\mathrm{std}} - 1$ by

$$\delta' = \frac{1+\delta}{1 + r_{\mathrm{std}}(1+\delta)} - 1 \tag{4}$$

$$\delta = \frac{1+\delta'}{1 - r_{\mathrm{std}}(1+\delta')} - 1 \tag{5}$$

In terms of this $\delta'$, the mass balance equations become

$$\frac{d}{dt}C = \sum_s F_s - C\sum_l L_l + r_{\mathrm{std}}C(\delta'_a + 1)\sum_l L_l(1-\alpha_l) \tag{6}$$

$$\frac{d}{dt}C\delta'_a = \sum_s \delta'_s F_s - C\delta'_a\sum_l \alpha_l L_l + C\sum_l L_l(1-\alpha_l) - r_{\mathrm{std}}C(\delta'_a + 1)\sum_l L_l(1-\alpha_l) \tag{7}$$

---

[1]There is not a single unique value of $r_{\mathrm{std}}$ in literature. Currently, $r_{\mathrm{std}} = 0.011180$ (Zhang and Li, 1990) is used by most measurement laboratories, while values of 0.011117 (Malinovsky et al., 2019) and 0.011125 (Fleisher et al., 2021) have also been reported recently. However, the true value of $r_{\mathrm{std}}$ impacts neither our formulation nor our results, as long as a single value is used consistently.





where $C = {}^{12}\mathrm{CH}_4 + {}^{13}\mathrm{CH}_4$ and $F_s = {}^{12}\mathrm{F}_s + {}^{13}\mathrm{F}_s$ are total methane moles and fluxes respectively. This reformulation of ${}^{13}\mathrm{CH}_4$

abundance in terms of total carbon is similar to that by Tans et al. (1993). In eq (6), if we consider the coefficients of any $L_l$,

then the second term supplies $C \sim 1800\,\mathrm{ppb}$, while the third term supplies $r_{\mathrm{std}}C(\delta'_a + 1)(1 - \alpha_l) \sim 0.086\,\mathrm{ppb}$, approximating

$r_{\mathrm{std}} = 0.01$, $\delta'_a = -0.05$ and $\alpha_l = 0.995$. In eq (7), with the same approximations, the coefficients of $L_l$ in the last three terms

are, respectively, $89.5\,\mathrm{ppb}$, $9\,\mathrm{ppb}$ and $0.086\,\mathrm{ppb}$. So in both equations, we ignore $r_{\mathrm{std}}C(\delta'_a + 1)\sum_l L_l(1 - \alpha_l)$, leading to

$$\frac{d}{dt}C \simeq \sum_s F_s - C\sum_l L_l \tag{8}$$

$$\frac{d}{dt}C\delta'_a \simeq \sum_s \delta'_s F_s - C\delta'_a \sum_l \alpha_l L_l + C\sum_l L_l(1 - \alpha_l) \tag{9}$$

In this formulation, the two tracers to be simulated are total $\mathrm{CH}_4$ (which is measured) and an artificial tracer $C\delta'_a$. All measurements of $\delta^{13}\mathrm{CH}_4$ are converted to $\delta'_a$ before assimilation. Note that the tracer $C\delta'_a$ does not have any surface flux of its own. There is "production" at the surface proportional to the the $\mathrm{CH}_4$ surface flux, and loss in the atmosphere. The loss reactions of eq (8) and (9) are coupled, and the loss of the tracers from time $t$ to $t + \delta t$ is calculated by solving the differential equation to

give

$$C(t + \delta t) = C(t)e^{-\delta t \sum_l L_l} \tag{10}$$

$$C\delta'(t + \delta t) = \left[C(t) + C\delta'(t)\right]e^{-\delta t \sum_l \alpha_l L_l} - C(t)e^{-\delta t \sum_l L_l} \tag{11}$$

## 2.2 Inversion framework

We use the TM5 4DVAR inversion framework (Meirink et al., 2008), which has been used to estimate surface fluxes of CO,

$\mathrm{CO}_2$ and $\mathrm{CH}_4$ (Hooghiemstra et al., 2011; Bergamaschi et al., 2013; Krol et al., 2013; Basu et al., 2013, 2014) in single-tracer inversions, as well as source-specific $\mathrm{CO}_2$ fluxes in multi-tracer inversions (Basu et al., 2020, 2016; Ma et al., 2021). At the heart of the framework is the TM5 offline tracer transport model (Krol et al., 2005) and its adjoint, driven by ECMWF ERA Interim reanalysis winds and run globally at $3° \times 2°$ with 25 vertical layers defined by sigma-pressure hybrid coordinates. Two tracers are simulated in TM5, total methane or $C$ of eq (8), and the artificial tracer $C\delta'_a$ of eq (9). Measurements of $\mathrm{CH}_4$

are directly compared to modeled values of $C$, while measurements of $\delta^{13}\mathrm{CH}_4$ are first converted to $\delta'_a$ and then to $C\delta'_a$ by multiplying with values of $\mathrm{CH}_4$ mole fractions measured in the same air samples.

TM5 4DVAR minimizes a cost-function $J$ as a function of surface fluxes $x$ by balancing fits to atmospheric observations $y$ with deviations from the prior fluxes $x_0$,

$$J(x) = \frac{1}{2}(Hx - y)^T R^{-1}(Hx - y) + \frac{1}{2}(x - x_0)B^{-1}(x - x_0) \tag{12}$$

where $H$ is the transport, chemistry and observation operator connecting surface fluxes with atmospheric measurements, and $R$ and $B$ are the error covariances of $Hx - y$ and prior fluxes respectively. Our formulation of $R$ contains both the analytical measurement uncertainty and a model representativeness error proportional to local tracer gradients (Meirink et al., 2008). For each source type (pyrogenic, fossil and microbial), the diagonal elements of $B$ per time step and lateral grid cell are



**Table 1.** Parameters for constructing the prior flux error covariance

| Source type | $f$ | $\lambda$ (km) | $\tau$ (months) |
|---|---|---|---|
| Microbial | 1.2 | 500 | 2 |
| Fossil | 1.5 | 700 | 6 |
| Pyrogenic | 1.0 | 300 | 1 |

proportional to the prior flux, or $f \times x_0$. Off-diagonal elements of $B$ are constructed assuming an exponential decay of the prior error correlation in space and time with source-specific scales $\lambda$ and $\tau$ respectively. The values of $f$, $\lambda$ and $\tau$ for the different source types are given in Table 1. While there is no unique way of specifying these parameters, our choices yield reasonable $1\sigma$ prior uncertainties on global total microbial, fossil and pyrogenic emissions of $\sim 25\,\mathrm{Tg\,yr^{-1}}$ ($\sim 7\%$), $\sim 30\,\mathrm{Tg\,yr^{-1}}$ ($\sim 17\%$) and $\sim 2\,\mathrm{Tg\,yr^{-1}}$ ($\sim 6\%$) respectively. The $1\sigma$ uncertainty on the annual global total $CH_4$ emission comes to $\sim 40\,\mathrm{Tg\,yr^{-1}}$ ($\sim 7\%$) with these choices. The cost function $J$ of eq (12) is minimized over 50 iterations by a conjugate gradient minimizer utilizing the Lanczos algorithm (Lanczos, 1950; Courtier et al., 1998).

In TM5 4DVAR, we calculate $J(x)$ of (12) with TM5 and $\partial J / \partial x$ with its adjoint. A traditional variational estimation would require us to run the forward and adjoint models multiple times over the entire period over which we want to estimate fluxes. However, these model runs require a significant amount of time, and iterations must be performed in succession. For example, at our $3° \times 2°$ resolution, TM5 simulates a decade in 8 hours. So to perform an inversion over two decades with 50 iterations (one iteration is one forward and one adjoint model run), it would take $8 \times 2 \times 2 \times 50 = 1600$ hours, or 67 days just for the model runs, not counting time spent in the computing queue. This is impractical given the need to do tests required of any new inversion system. Therefore, we split up our target period into several inversions that were run in parallel as shown in Figure 2(a). A single forward run from 1984 to 2017 produced initial $C$ and $C\delta'_a$ fields for all inversions. This forward run was identical to scenario "C_WL+" of Lan et al. (2021) and matched the long term atmospheric $CH_4$ and $\delta^{13}CH_4$ trends over that period. Six five-year inversions were run simultaneously with two years of overlap (red bars) between inversions, starting in 1997, 2000, 2003, 2006, 2009 and 2012. After all six inversions finished, the fluxes from the middle three-year period of each inversion (blue bars) were considered for analysis. For simulating prior and posterior mole fractions, fluxes from the non-overlapping periods (1997 – 2001, 2001 – 2004, 2004 – 2007 ... 2013 – 2017) were stitched together and a single forward run was done with those fluxes.

## 2.3 Prior fluxes and $\delta^{13}CH_4$ source signatures

The prior fluxes and their $\delta^{13}CH_4$ source signatures for the different categories of methane emissions are described in detail as "scenario C_WL+" in Lan et al. (2021). Briefly, the prior fluxes are based on bottom-up emission estimates with adjustments to match global atmospheric $CH_4$ increases and to satisfy the global mass balance of $\delta^{13}CH_4$ over 1984–2017. For biomass burning or pyrogenic emissions, we use the Global Fire Emission Database (GFED) 4.1s for 1997-2016 (van der Werf et al., 2017) and estimates from the Reanalysis of Tropospheric chemical composition (RETRO) project before 1997 (Schultz et al.,





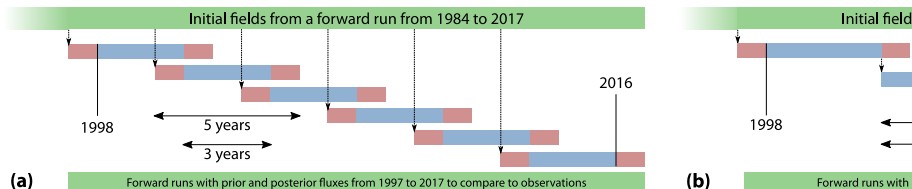

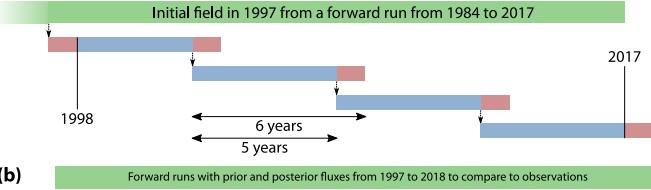

**Figure 2.** A schematic of the time splitting of our inversions. Red bars denote spin up and spin down periods, and blue bars denote periods from which fluxes were considered in our analysis. Schematic (a) denotes the time splitting used in most of our inversions as described in § 2.2, whereas schematic (b) denotes the time splitting used specifically with climatological priors as described in § 3.4. In splitting scheme (a), each inversion spans five years and the entire time span is covered with six inversions running simultaneously, starting from initial fields produced by a 1984–2017 model run with prior fluxes. In splitting scheme (b), each inversion spans six years and the entire span is covered with four inversions. However, except for the 1997–2003 inversion, all other inversions are started from the optimized mole fraction fields at the end of year 5 of the previous inversion, and therefore the inversions cannot be run in parallel.

2008). Other anthropogenic emissions are based on the EDGAR 4.3.2 inventory (Janssens-Maenhout et al., 2019). We use natural fossil emissions reported by Etiope et al. (2019). Emission estimates from wild animals and termites are adopted from Bergamaschi et al. (2007). Wetland emissions and upland soil consumption of methane are estimated by a process-based model (Zhuang et al., 2004; Liu et al., 2020), after which the soil sink is modeled as a $1^{st}$ order loss process as explained in § 2.4.

The $\delta^{13}CH_4$ source signatures used in our study are mainly spatially resolved maps based on the Global $\delta^{13}CH_4$ Source Signature Inventory 2020 for coal, oil and gas (ONG), biomass and biofuel burning, ruminant and wild animal sources (Sherwood et al., 2021; Lan et al., 2021), spatial maps for geological seeps (Etiope et al., 2019) and wetland sources (Ganesan et al., 2018). Globally averaged values are used for waste, landfills, termites, rice, and other energy and industry, given insufficient data to develop spatial distributions for their $\delta^{13}CH_4$ signatures (Lan et al., 2021).

The sum of the bottom-up methane emission estimates described above is not consistent with top-down estimates of global total emissions based on observed atmospheric $CH_4$ growth and estimated loss, which requires a $46\,\mathrm{Tg\,yr^{-1}}$ increase in the annual global emission in 2016 compared to the 1999–2006 quasi-stable period. In addition, the $\delta^{13}CH_4$ mass balance requires $167\,\mathrm{Tg\,yr^{-1}}$ emissions from fossil sources (including natural geological seeps) to be consistent with modeled sinks and the $\delta^{13}CH_4$ source signatures described above. Therefore, we (i) scale the ONG emissions from EDGAR 4.3.2 uniformly using

annual scaling factors to reach a total of $167\,\mathrm{Tg\,yr^{-1}}$ from all fossil sources, (ii) impose a linear trend on wetland emissions to achieve an increase of $46\,\mathrm{Tg\,yr^{-1}}$ in total 2016 emissions compared to 1999–2006, and (iii) adjusted emissions from agricultural and wastes sectors to match the year on year global $CH_4$ growth rate derived from marine boundary layer observations (Dlugokencky et al., 2011). This ensures that our global $CH_4$ and $^{13}CH_4$ budgets approximate the long term trends in atmospheric $CH_4$ and $\delta^{13}CH_4$ over 1984–2017.





**Table 2.** Fractionation parameters for CH$_4$ loss

| Loss reaction | C | D (K) | Reference |
|---|---|---|---|
| Loss to OH | 1.0039 | 0.00 | Saueressig et al. (2001) |
| Loss to Cl | 1.0430 | 6.46 | Saueressig et al. (2001) |
| Loss to O($^1$D) | 1.0130 | 0.00 | Saueressig et al. (2001) |
| Soil sink | 1.0215 | 0.00 | King et al. (1989) |

## 2.4 Methane loss mechanisms and fractionation

Atmospheric methane has four loss mechanisms, atmospheric oxidation by OH and Cl throughout the atmosphere, destruction by O($^1$D) in the stratosphere, and surface uptake by upland soils. In all our inversions, these sinks are prescribed and not optimized. Monthly climatological CH$_4$ loss rates in the stratosphere due to OH, Cl and O($^1$D) were constructed from a run of the ECHAM5/MESSy1 chemistry transport model (Steil et al., 2003; Jöckel et al., 2006). Loss due to tropospheric Cl is simulated using a recent model-derived estimate of tropospheric Cl (Hossaini et al., 2016). For tropospheric OH, we use the monthly OH climatology of Spivakovsky et al. (2000) after scaling by 0.9 to match the declining atmospheric abundance of methyl chloroform in the early 2000s (Montzka et al., 2011).

In most CH$_4$ inversions, upland soil sinks are folded into the net wetland flux. However, the soil sink fractionates strongly between $^{12}$CH$_4$ and $^{13}$CH$_4$ (King et al., 1989), and therefore we keep it separate from wetland fluxes. We model the soil sink as a first order reaction at the surface, in which the loss rates of $^{12}$CH$_4$ and $^{13}$CH$_4$ in the lowest model layer are $d^{12}C/dt = -k_{ss}F_{ss}\,^{12}C$ and $d^{13}C/dt = -\alpha_{ss}k_{ss}F_{ss}\,^{13}C$ respectively. $F_{ss}$ is the prior soil sink map from the TEM land surface model, and $k_{ss}$ is an arbitrary constant tuned to a value such that in a forward run with prior fluxes, the global total soil sink matches the prior total.

The fractionation between $^{12}$C and $^{13}$C for each of the loss reactions is modeled as $k_{12}/k_{13} = 1/\alpha = Ce^{D/T}$ (Saueressig et al., 2001), where $T$ is the air temperature in Kelvin. The soil sink fractionation is cast in a similar form for convenience. Coefficients $C$ and $D$ we used are tabulated in Table 2.

## 2.5 CH$_4$ and $\delta^{13}$CH$_4$ measurements

To maximize the spatiotemporal coverage of in-situ CH$_4$ and $\delta^{13}$CH$_4$ data, we have developed a new database by harmonizing measurements from NOAA/INSTAAR with those from 30 other laboratories around the world (Lan et al., 2021). All CH$_4$ data have been quality checked and converted to a common CH$_4$ scale, namely the World Meteorological Organization (WMO) X2004A scale maintained at NOAA's Global Monitoring Laboratory (Dlugokencky et al., 2005). For data not on the WMO X2004A scale, we applied lab-specific scale multipliers estimated based on (i) comparisons of measurements of common air samples during the WMO/IAEA Round Robin Comparison Experiment (Crotwell et al., 2020), and (ii) comparisons of co-located atmospheric measurements made by NOAA and other laboratories. We constructed the uncertainty on the assimi-





lated $CH_4$ measurements from a combination of (i) measurement repeatability of a single sample (hereafter called the single measurement precision), (ii) lab-specific long-term reproducibility based on analyzer type and sampling frequency reported in literature, and (iii) each lab's realization of the calibration scale. If a scale conversion was needed to bring measurements onto the WMO X2004A scale, the mole fraction uncertainty due to the scale multiplier uncertainty was added in quadrature. The final uncertainties are typically less than $9\,\mathrm{ppb}$ for all $CH_4$ measurements.

We used $\delta^{13}CH_4$ data from the Institute for Arctic and Alpine Research (INSTAAR) as well as other isotope laboratories making precise measurements of atmospheric methane with isotope ratio mass spectrometers. The INSTAAR $\delta^{13}CH_4$ data were measured in a subset of air samples collected from NOAA's Global Greenhouse Gas Reference Network (GGGRN). Because different labs have independent ties to primary reference materials which do not agree, we calculated offsets to bring the $\delta^{13}CH_4$ data onto the INSTAAR realization of the Vienna Pee Dee Belemnite (VPDB) scale (Miller et al., 2002). These

offsets were based on measurements of cylinders, flasks filled from cylinders, or co-located sample data, and are all described in Umezawa et al. (2018). When there was not a direct comparison, e.g., between INSTAAR and TU, or INSTAAR and NIPR, we used comparisons between each of these labs and the Institute for Marine and Atmospheric research Utrecht (IMAU). Each comparison had an uncertainty associated with it, which were combined in quadrature to account for uncertainty in the offset correction. The total uncertainty on assimilated $\delta^{13}CH_4$ measurements was typically less than $0.15\,\text{‰}$. The final database of

assimilated $CH_4$ and $\delta^{13}CH_4$ measurements is available at https://doi.org/10.15138/64w0-0g71.

   With the following exceptions, we assimilate all the observations from this database including marine boundary layer sites, surface and tower sites over continents (Andrews et al., 2014), and vertical profiles from routine aircraft measurements (Sweeney et al., 2015). Intermittent aircraft profiles such as from the HIPPO (Wofsy, 2011) and ATom (Thompson et al., 2022) campaigns are not assimilated. $CH_4$ data from flasks taken aboard routine flights between Japan and Australia as part of the

CONTRAIL program have been assimilated (Machida et al., 2008; Matsueda et al., 2015). A subset of the CONTRAIL flasks were also analyzed for $\delta^{13}CH_4$ (Umezawa et al., 2012), which were not assimilated. For continental tower sites with multiple intake heights, only data from the highest intake are considered in inversions to minimize local influence. For sites with continuous $CH_4$ analyzers, the $CH_4$ data are averaged hourly and only hourly averages between 11:00 and 16:00 local solar time are assimilated; these are the times when planetary boundary layer heights are likely to be best-represented by transport

models. For continuous $CH_4$ analyzers on mountain tops, we only assimilate hourly averages between 00:00 to 05:00 local solar time to avoid possible up-slope contamination. Site-specific statistical filtering based on a non-parametric curve fitting routine (Thoning et al., 1989) is further applied, with the exception of vertical profiles, to remove large outliers with potential local or other contamination. The number of $CH_4$ and $\delta^{13}CH_4$ measurements assimilated each year is summarized in Table 3, and their locations are plotted in Figure 3.

**2.6   Uncertainty estimation and sensitivity tests**

The uncertainty of surface emission estimates is a combination of random and systematic uncertainties. Random uncertainties are associated with those components of the inversion system whose variations are assumed to be zero on average. In the formulation of the cost function (12), the prior flux $x_0$ is assumed to have a probability density function (PDF) centered on





**Table 3.** The number of $CH_4$ and $\delta^{13}CH_4$ observations assimilated in our inversions, broken down by year.

| Year | $CH_4$ | $\delta^{13}CH_4$ | Year | $CH_4$ | $\delta^{13}CH_4$ | Year | $CH_4$ | $\delta^{13}CH_4$ |
|------|--------|-------------------|------|--------|-------------------|------|--------|-------------------|
| 1997 | 9075 | 0 | 2004 | 24669 | 1178 | 2011 | 66307 | 1914 |
| 1998 | 9236 | 457 | 2005 | 36077 | 742 | 2012 | 74957 | 1842 |
| 1999 | 9981 | 371 | 2006 | 36707 | 1163 | 2013 | 70785 | 1592 |
| 2000 | 33514 | 537 | 2007 | 44056 | 1042 | 2014 | 81433 | 2171 |
| 2001 | 16514 | 256 | 2008 | 51138 | 990 | 2015 | 84900 | 2576 |
| 2002 | 19497 | 925 | 2009 | 53243 | 1875 | 2016 | 81126 | 2941 |
| 2003 | 20191 | 1070 | 2010 | 66930 | 1413 | 2017 | 57977 | 2337 |

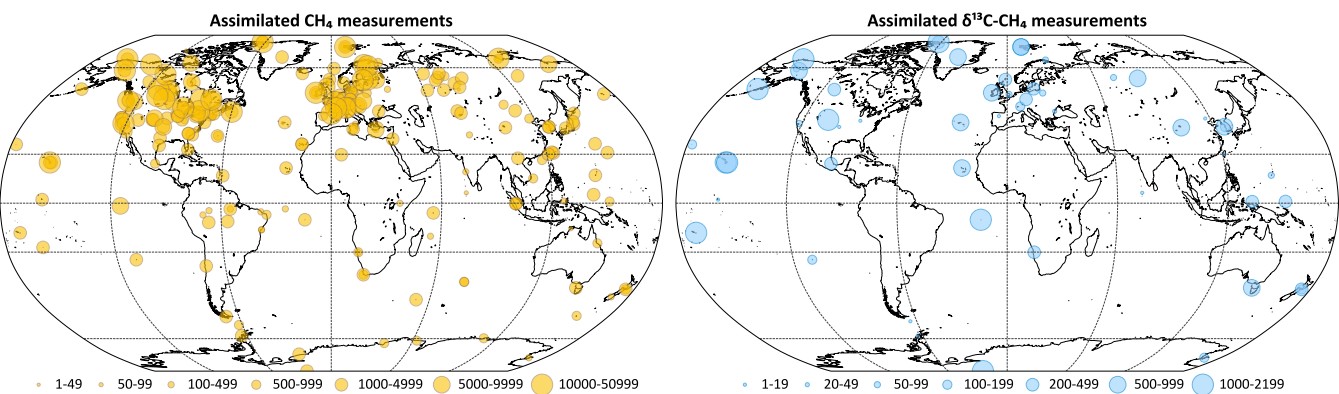

**Figure 3.** Locations of assimilated $CH_4$ and $\delta^{13}CH_4$ measurements. The symbol sizes represent the number of measurements between 1997–2017 assimilated from each location. Overlapping symbols over some of the locations are due to multiple agencies measuring at those locations.

the true flux with variance around the truth given by the prior covariance matrix $B$. Similarly, the model-observation mismatch
$Hx - y$ is assumed to have a PDF centered around the mismatch between the true atmospheric mole fraction and true fluxes propagated through an unbiased transport model, with variance around this mean given by $R$. The random uncertainty in the optimal estimate is given by $\hat{B}$, where

$$\hat{B}^{-1} = \frac{\partial^2 J}{\partial x^2} = H^T R^{-1} H + B^{-1} \tag{13}$$

Variational inversion systems such as TM5 4DVAR can construct a low rank approximation of $\hat{B}$ during the optimization.
However, for large state vectors the $\hat{B}$ thus constructed is an overestimation of the true posterior uncertainty (Meirink et al., 2008; Bousserez et al., 2015). We therefore construct an estimate of $\hat{B}$ by performing an ensemble of 100 independent inversions for each of the 5-year inversions of Figure 2(a), with prior fluxes and observations perturbed according to the covariances specified by $B$ and $R$ respectively. With 100 ensemble members, our estimate of $\hat{B}$ is expected to be within $10\%$ of the exact



analytical solution for $\hat{B}$ (Bousserez et al., 2015). Furthermore, our ensemble of inversions allows us to compute any posterior
covariance and correlation between estimated fluxes, such as between large regions or different $CH_4$ source types.

Systematic uncertainties are associated with aspects of the inversion system that are assumed fixed and perfectly known
in principle, but might in fact be biased in practice. In our inversion system, such aspects include, but are not limited to,
atmospheric transport and chemistry, isotope source signatures, and the wetland inundation maps used to construct the prior
wetland emissions. Because the posterior covariance estimate does not include systematic errors, we explore the impact of such
errors by performing inversions with different realizations of those potentially biased inputs in the following sensitivity tests.

### 2.6.1 Tropospheric chlorine

The magnitude and distribution of the Cl sink in the troposphere is uncertain, with estimates as high as $13\,\mathrm{Tg\,yr^{-1}}$ to
$37\,\mathrm{Tg\,yr^{-1}}$ based primarily on southern hemisphere background observations (Allan et al., 2007). However, more recent stud-
ies have found a more limited role of tropospheric Cl as a methane oxidant (Gromov et al., 2018). Consequently, most $CH_4$
inverse models neglect tropospheric Cl as a methane oxidant. However, due to the strong isotopic fractionation in the $CH_4 + Cl$
reaction, Cl plays an important role in determining atmospheric $\delta^{13}CH_4$ (Strode et al., 2020; Lan et al., 2021). It is therefore
important to test the sensitivity of our conclusions to the imposed tropospheric Cl sink within the range of realism. The Cl
estimate by Hossaini et al. (2016) we use in this study is on the higher side of the range posited by Gromov et al. (2018). We
perform an inversion with the tropospheric Cl field reported by Wang et al. (2021) as an alternative lower specification. In order
to keep the global $CH_4$ lifetime unchanged between the two scenarios of tropospheric Cl, we scale the tropospheric OH field
by 0.9 and 0.92 respectively when we use the Cl fields of Hossaini et al. (2016) and Wang et al. (2021). Since the two scenarios
lead to slightly different sink fractionation in the atmosphere, prior ONG and ruminant fluxes are adjusted to match the long
term atmospheric $\delta^{13}CH_4$ trend for both cases.

### 2.6.2 OH fractionation

We use chemical fractionation factors reported by Saueressig et al. (2001) since they provide factors for all atmospheric sink
processes from a consistent set of laboratory measurements. While these are the most recent and generally accepted, for $CH_4$
oxidation by OH another set of coefficients $C = 1.0054, D = 0$ have previously been reported by Cantrell et al. (1990). To the
best of our knowledge, this earlier result has not been refuted in the literature, nor is there any independent evidence supporting
one set of coefficients over another. Instead, the most recent evaluation of atmospheric reaction rates (Burkholder et al., 2019)
recommends using the Saueressig et al. (2001) rates with increased uncertainty in the OH fractionation to include Cantrell et al.
(1990) as a possibility. Since the sink fractionation plays a significant role in determining atmospheric $\delta^{13}CH_4$, we perform
an additional inversion with the OH fractionation of Cantrell et al. (1990) to gauge its impact. Since the two OH fractionation
factors lead to different sink fractionation in the atmosphere, prior ONG and ruminant fluxes are adjusted to match the long
term atmospheric $\delta^{13}CH_4$ trend for both cases.





### 2.6.3 $\delta^{13}CH_4$ source signatures

In principle, it is possible to estimate both $CH_4$ fluxes and $\delta^{13}CH_4$ source signatures in a dual tracer inversion (Thanwerdas et al., 2021). However, this makes the problem non-linear and the inversion convergence slow. It is also difficult to construct a prior covariance for $\delta^{13}CH_4$ source signatures since much of the uncertainty stems from extrapolating a limited number of $\delta^{13}CH_4$ signature measurements to the entire domain of $CH_4$ sources, resulting in errors that are systematic and non-Gaussian. Therefore, we explore the impact of $\delta^{13}CH_4$ signature uncertainty on our results by running inversions with alternate specifications of $\delta^{13}CH_4$ signature maps as follows.

Source signature maps for biomass burning were calculated by multiplying C3 and C4 signatures of $-26.7\,‰$ and $-12.5\,‰$ respectively (Cerling et al., 1998) with the C3/C4 fraction for each $1° \times 1°$ latitude/longitude grid cell (Lan et al., 2021). For ruminants and wild animals, C3 and C4 signatures were taken to be $-54.5\,‰$ and $-67.8\,‰$ respectively from the Global $\delta^{13}CH_4$ Source Signature Inventory 2020 (Sherwood et al., 2021; Lan et al., 2021). In this way, the C3/C4 vegetation distribution determines the source signatures of both biomass burning and ruminant emissions. Our default inversion averages the C3/C4 distributions of Still et al. (2003) and its modified version as used by Randerson et al. (2012). To explore the uncertainty from the assumed C3/C4 map, we perform two additional inversions with $\delta^{13}CH_4$ source signature maps derived separately from the two individual C3/C4 distributions. In addition, country-level ruminant emission signatures were compiled by Chang et al. (2019), including their temporal changes due to shifting ruminant diet and due to the downward trend in atmospheric $\delta^{13}CO_2$ that is photosynthesized by the vegetation. We use the ruminant $CH_4$ source signatures of Chang et al. (2019) in a third inversion. The three instances of source signatures related to the C3/C4 distribution described here were significantly different, requiring us to adjust the prior flux apportionment to meet our goal of matching long-term $CH_4$ and $\delta^{13}CH_4$ trends. Specifically, we changed the prior fossil $CH_4$ emissions from the default of $167\,\mathrm{Tg\,yr^{-1}}$ to $158\,\mathrm{Tg\,yr^{-1}}$ for the inversions using $\delta^{13}CH_4$ signatures derived from Still et al. (2003) and Chang et al. (2019). For the inversion using $\delta^{13}CH_4$ signatures derived from Randerson et al. (2012), we adjusted the prior fossil emission to $175\,\mathrm{Tg\,yr^{-1}}$. In all cases, this was achieved by globally scaling the ONG and ruminant emissions to achieve long-term $CH_4$ and $^{13}CH_4$ mass balance.

For the global maps of ONG and coal emission signatures, our default inversion assumes time-invariant maps over the study period. However, considering the rapid development of the US shale gas production and a shift in production from conventional to shale gas in the past decades, we estimate that the mean US ONG signature (production-weighted mean of shale and conventional gas) increased by $2.7\,‰$ from 2006 to 2016 (Lan et al., 2021). We incorporate this in an alternate specification of fossil $CH_4$ source signatures and perform an inversion with this new map.

Finally, our default inversion setup uses the latitude-based source signature specification of Ganesan et al. (2018) for wetland emissions. Over the past several years we have implemented carbon isotopes in the TEM land surface model (Zhuang et al., 2004), making it possible to derive process-based $\delta^{13}CH_4$ wetland source signatures consistent with wetland emissions (IsoTEM, Oh et al., 2021). We perform an inversion with wetland source signatures from the IsoTEM model as an alternative to our default wetland source signatures.





### 2.6.4 Wetland inundation extent

Wetland inundation extent is a leading driver of uncertainty in bottom-up estimates of wetland $CH_4$ emissions, and therefore in
the global $CH_4$ budget. We explore this uncertainty by performing inversions with prior wetland $CH_4$ fluxes derived from the
TEM model (Zhuang et al., 2004) driven by two different inundation maps. Our default setup uses a time-varying or dynamic
inundation map based on the satellite-based Surface WAter Microwave Product Series (SWAMPS, Schroeder et al., 2015)
combined with the Global Lakes and Wetlands Dataset (GLWD, Lehner and Döll, 2004; Poulter et al., 2017). In addition, we
also drive the TEM model with the static inundation map of Matthews and Fung (1987), in which case meteorology is the only
source of seasonal and inter-annual variation of prior wetland emissions. These two inundation maps produce significantly
different atmospheric $CH_4$ and $\delta^{13}CH_4$ gradients (scenarios "C_WL+" and "Q_static_WL" of Lan et al., 2021) in a forward
run, and therefore serve as a robust test of our inversion results with different inundation extents.

### 2.6.5 Initial $\delta^{13}CH_4$ gradients

Large scale gradients of atmospheric $\delta^{13}CH_4$ take significantly longer to respond to changes in emissions compared to gradients
of $CH_4$ (Tans, 1997), requiring multi-decade spin-ups for models trying to simulate atmospheric $\delta^{13}CH_4$ (Lan et al., 2021).
Inverse models, on the other hand, take significantly less time to be spun-up since fluxes during the spin-up period are modified
to fit observed atmospheric $\delta^{13}CH_4$. The exact spin-up duration required depends on the accuracy of the initial modeled $\delta^{13}CH_4$
gradients and the inversion setup. To test if a one year spin-up for our inversions as depicted in Figure 2(a) is sufficient,
we perform two additional inversions with different starting $\delta^{13}CH_4$ large-scale gradients. Specifically, of the flux scenarios
simulated by Lan et al. (2021), we choose scenarios "H_mean_sig" and "Q_static_WL", which produced the flattest and
steepest north-south gradients in $\delta^{13}CH_4$ respectively (see Lan et al., 2021, Figure 5). We perform inversions starting from
$CH_4$ and $\delta^{13}CH_4$ fields provided by forward simulations of those scenarios at each of the starting points in Figure 2(a). The
resultant spread in fluxes provides an estimate of the sensitivity of our setup to erroneous initial $\delta^{13}CH_4$ gradients.

## 3 Results

### 3.1 Fit to atmospheric $CH_4$ and $CH_4 + \delta^{13}CH_4$ data

Both the $CH_4$-only and the $CH_4 + \delta^{13}CH_4$ inversions fit the atmospheric $CH_4$ data, while only the latter is consistent with
atmospheric $\delta^{13}CH_4$ data. This is demonstrated both at surface sites from which data were assimilated, as well as data from
aircraft campaigns that were withheld for validation. Figure 4 shows that both inversions fit the observed $CH_4$ time series at
three NOAA baseline observatories. However, despite starting from realistic atmospheric $CH_4$ and $\delta^{13}CH_4$ fields, the $CH_4$-only
inversion moves progressively farther from observed $\delta^{13}CH_4$ with time at those same locations, demonstrated in Figure 5. Only
the $CH_4 + \delta^{13}CH_4$ inversion fits both atmospheric $CH_4$ and $\delta^{13}CH_4$ data. This is also demonstrated in Figure 6, which compares
modeled $\delta^{13}CH_4$ to $\delta^{13}CH_4$ measured by the HIPPO and ATom aircraft campaigns, and from regular flights between Japan and
Oceania as part of the CONTRAIL program. ATom and HIPPO campaigns sampled primarily background air over the oceans



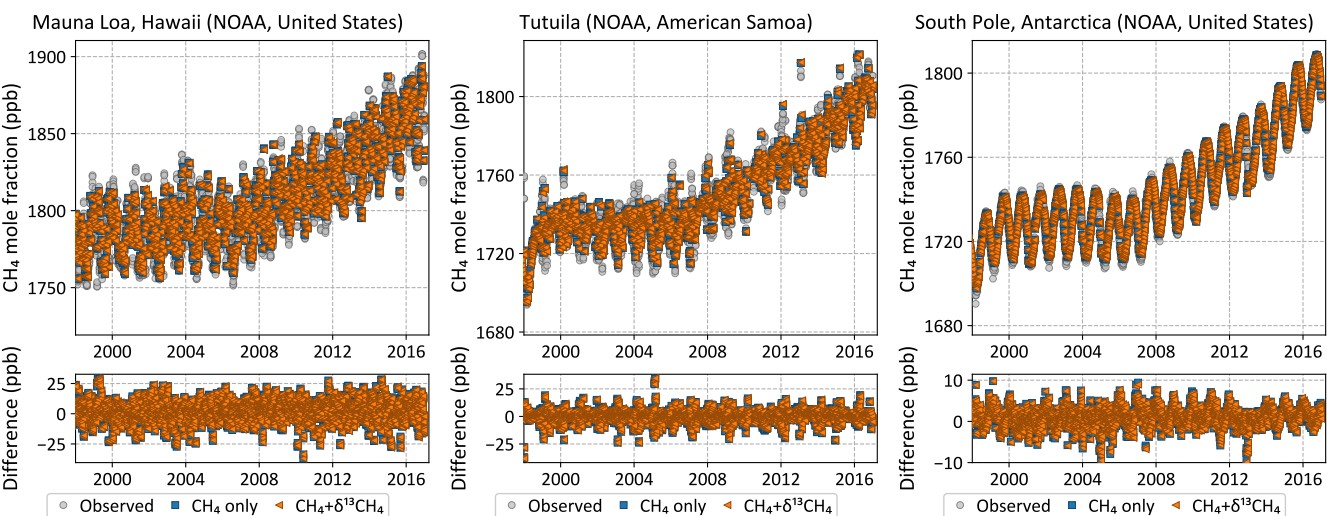

**Figure 4.** Observed (grey circles) and posterior modeled (colored symbols) $CH_4$ time series at three NOAA baseline observatories. Both inversions with and without $\delta^{13}CH_4$ data fit the $CH_4$ data equally well.

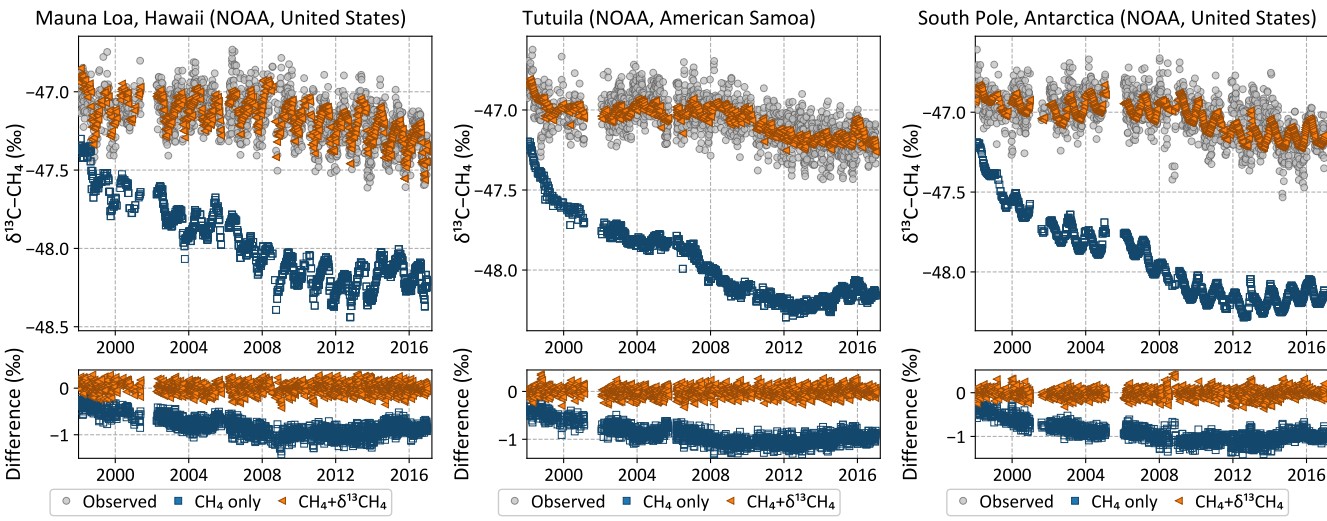

**Figure 5.** Observed (grey circles) and posterior modeled (colored symbols) $\delta^{13}CH_4$ time series at three NOAA baseline observatories. The inversion with $\delta^{13}CH_4$ data fit the observations throughout the inversion period, but the inversion without $\delta^{13}CH_4$ data – a traditional $CH_4$ inversion – drifts away from the observations with time. Note that both inversions were started with the same $CH_4$ and $\delta^{13}CH_4$ fields in 1997, but by the time $\delta^{13}CH_4$ data were available in mid-1998 they had already drifted apart, leading to the apparent initial offset in the plots above.





**Figure 6.** Modeled minus observed $\delta^{13}CH_4$ as a function of latitude and altitude from the ATom (top), HIPPO (middle) and CONTRAIL (bottom) aircraft campaigns. Mismatches are shown for the prior flux, the $CH_4$-only inversion, and the dual tracer $CH_4 + \delta^{13}CH_4$ inversion. Altitudes have been binned in 1 km bins, while latitudes have been binned either in 10° (ATom, HIPPO) or 5° (CONTRAIL) bins. The rightmost panels show the number of samples averaged per bin.




at multiple latitudes and altitudes, and neither CH$_4$ nor $\delta^{13}$CH$_4$ data from those campaigns were assimilated. CONTRAIL

primarily sampled the marine background at multiple altitudes as well, except for a small number of samples taken during

takeoff and touchdown in Japan. CH$_4$ flask samples from CONTRAIL were assimilated in both inversions, but their $\delta^{13}$CH$_4$

measurements were not assimilated. The CH$_4$-only inversion compares far less favorably to the $\delta^{13}$CH$_4$ measurements than the

joint inversion. Therefore, it is reasonable to conclude that our CH$_4$-only inversion, and very likely most traditional CH$_4$-only

inversions, do not yield a CH$_4$ emission distribution consistent with atmospheric $\delta^{13}$CH$_4$ observations. We therefore expect our

CH$_4$ + $\delta^{13}$CH$_4$ inversion to provide more accurate emission estimates and source partitioning than our CH$_4$-only inversion.

## 3.2   Large scale fluxes from CH$_4$ and CH$_4$ + $\delta^{13}$CH$_4$ inversions

**Figure 7.** Total and source-specific annual emissions of CH$_4$ globally and from three latitudinal bands. "Tropics" in this context refers to the region between 23.5 °N and 23.5 °S, while the northern and southern extra-tropics are to the north and south respectively. The shaded regions denote $2\sigma$ prior and posterior error bars.





**Table 4.** Annual averages of $CH_4$ emissions between 1999–2016 and their $2\sigma$ uncertainties shown in Figure 7, in $\mathrm{Tg\,yr^{-1}}$. The "$CH_4$ only" and "$CH_4 + \delta^{13}CH_4$" inversions of Figure 7 have been abbreviated to "$CH_4$" and "Joint" respectively below. Note that (a) the $2\sigma$ uncertainties tabulated are the averages of the uncertainties across 18 years, *not* the uncertainties on the 18 year average emissions, and (b) an uncertainty of zero below is due to rounding, and not because the uncertainty is exactly zero.

| Source type | Total | | | Microbial | | | Fossil | | | Pyrogenic | | |
|---|---|---|---|---|---|---|---|---|---|---|---|---|
| Inversion | Prior | $CH_4$ | Joint | Prior | $CH_4$ | Joint | Prior | $CH_4$ | Joint | Prior | $CH_4$ | Joint |
| Globe | $577\pm79$ | $576\pm4$ | $576\pm4$ | $383\pm52$ | $407\pm13$ | $374\pm13$ | $167\pm59$ | $141\pm13$ | $173\pm13$ | $28\pm3$ | $28\pm3$ | $30\pm3$ |
| Northern Extra-Tropics | $318\pm68$ | $273\pm10$ | $274\pm10$ | $188\pm39$ | $164\pm14$ | $168\pm14$ | $120\pm55$ | $99\pm13$ | $96\pm13$ | $10\pm2$ | $10\pm2$ | $10\pm2$ |
| Tropics | $221\pm31$ | $271\pm11$ | $269\pm11$ | $163\pm27$ | $216\pm13$ | $184\pm13$ | $41\pm15$ | $38\pm10$ | $66\pm10$ | $17\pm3$ | $16\pm2$ | $19\pm2$ |
| Southern Extra-Tropics | $38\pm12$ | $32\pm4$ | $32\pm4$ | $32\pm11$ | $27\pm5$ | $22\pm5$ | $6\pm4$ | $4\pm3$ | $10\pm3$ | $1\pm0$ | $1\pm0$ | $1\pm0$ |

The top row of Figure 7 shows the global total annual emissions from two inversions, a $CH_4$-only or "traditional" methane inversion without $\delta^{13}CH_4$ data, and a joint $CH_4 + \delta^{13}CH_4$ inversion developed in this work. The shaded regions in Figure 7 denote $2\sigma$ random errors derived from 100-member Monte Carlo ensembles of inversions described in § 2.6. Annual averages of the emissions and random errors are summarized in Table 4. The global total emission from all categories is unaffected by the addition of $\delta^{13}CH_4$ data, since $\delta^{13}CH_4$ does not place any additional constraint on the total $CH_4$ emission. However, the partitioning between microbial and fossil sources is changed significantly with the addition of $\delta^{13}CH_4$ data. Based on comparison to atmospheric data as noted in § 3.1, we expect the source partitioning from our $CH_4 + \delta^{13}CH_4$ inversion to be more accurate compared to our traditional $CH_4$-only inversion.

Figure 7 also shows the total and source-disaggregated $CH_4$ emissions from our $CH_4$ and $CH_4 + \delta^{13}CH_4$ inversions over three latitude bands, where the Tropics are bounded between $23.5\,°\mathrm{S}$ and $23.5\,°\mathrm{N}$. Relative to the prior, tropical (extra-tropical) total emissions are adjusted upward (downward) by both inversions, and there is little sensitivity of the tropical versus extra-tropical partitioning on the assimilation of $\delta^{13}CH_4$ data. In the northern extra-tropics, the partitioning of $CH_4$ emissions between the different source types does not change significantly with the addition of $\delta^{13}CH_4$ data. However, in the Tropics the inversion with $\delta^{13}CH_4$ data shows significantly higher fossil (and lower microbial) emissions than the inversion without $\delta^{13}CH_4$ data. Fossil $CH_4$ emissions in the southern extra-tropics are significantly different for most years in the presence of $\delta^{13}CH_4$ data, but similarly significant differences do not exist for the other source types. Finally, our estimate of pyrogenic emissions does not change significantly in Figure 7 in the presence and absence of $\delta^{13}CH_4$.

### 3.3 Systematic errors in emission estimates

As explained in § 2.6, we estimate possible biases in our flux estimates by running the inversion with different choices of non-optimized input. The spread in annual emissions due to alternate specifications of atmospheric chemistry (tropospheric chlorine of § 2.6.1 and OH fractionation of § 2.6.2) is shown in Figure 8. Analogous spreads due to different specifications of $\delta^{13}CH_4$ source signatures (§ 2.6.3), wetland inundation maps (§ 2.6.4) and initial atmospheric $\delta^{13}CH_4$ fields (§ 2.6.5) are shown in Figure 9. Note that the Y-axis ranges in Figures 8 and 9 are different. The average spread in annual emissions from

350 different latitude bands and source types are summarized in Table 5 for each sensitivity test. The average of the annual posterior uncertainties as depicted in Figure 7 are also provided in Table 5 as "MC-derived ($2\sigma$)" for reference, with the caveat that $2\sigma$ uncertainties are not directly comparable to the range across a few inversions.

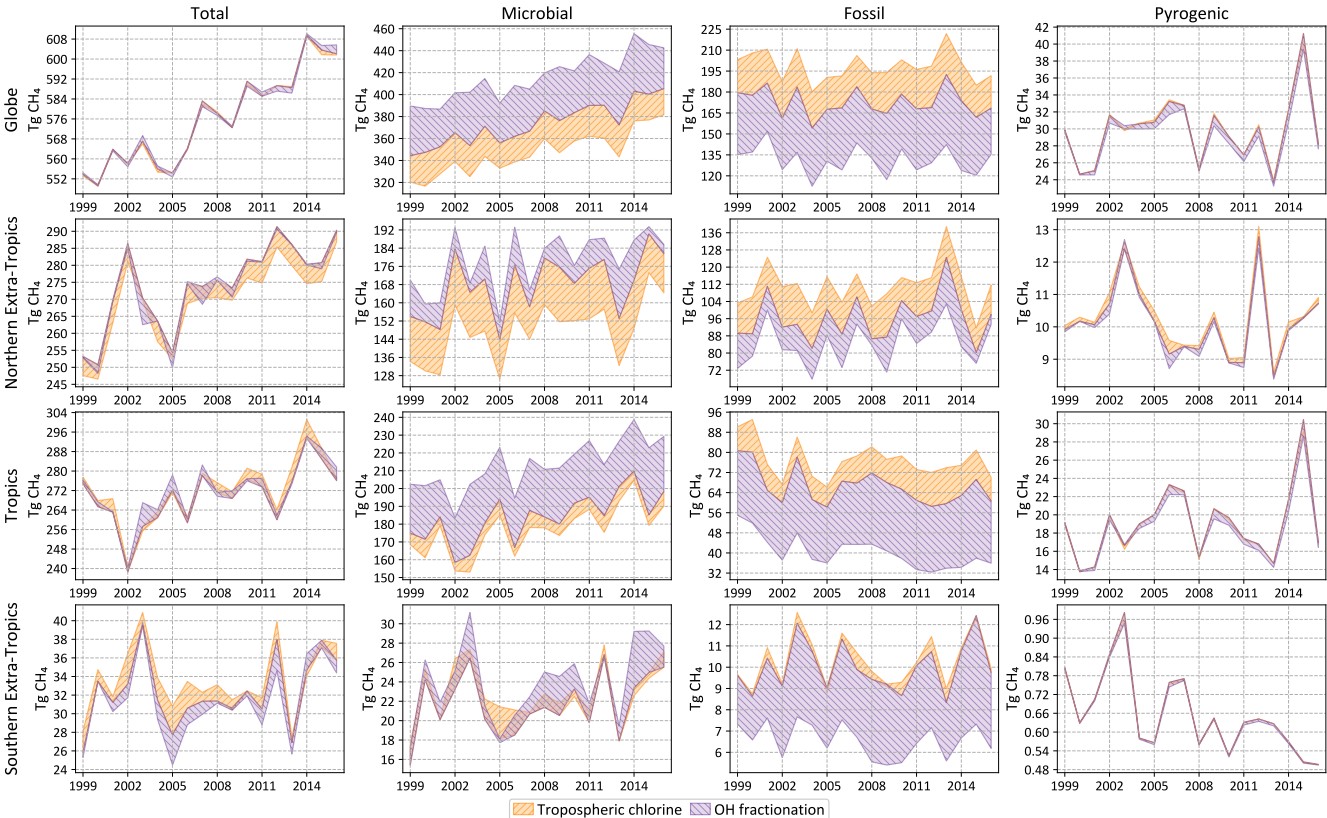

**Figure 8.** Total and source-specific annual emissions of CH$_4$ globally and from three latitudinal bands as in Figure 7. The shaded regions denote the spread (max to min) of annual emissions from sensitivity tests described in § 2.6.1 and § 2.6.2.

Most of the sensitivity tests have little impact on the global total CH$_4$ emission, and the spread in the total CH$_4$ emission from different latitude bands is generally smaller than the posterior uncertainty of our base inversion. However, by far the

355 largest source of error in partitioning the total emission into fossil and microbial sources comes from our representation of atmospheric chemistry, namely the distribution of tropospheric chlorine and the kinetic isotope effect of CH$_4$ destruction by OH. Unless the uncertainty in these two factors can be reduced, our ability to use $\delta^{13}$CH$_4$ measurements to partition different source types will be seriously hampered. The uncertainty arising from our limited knowledge of $\delta^{13}$CH$_4$ source signatures, to the extent represented by the different signature maps used, is lower than the uncertainty due to atmospheric chemistry. Lastly,

the uncertainty due to an incorrect specification of the initial atmospheric $\delta^{13}$CH$_4$ field is minimal, in line with our expectation that an inversion will rapidly correct for it by adjusting emissions during its spin-up period. However, we note here that the





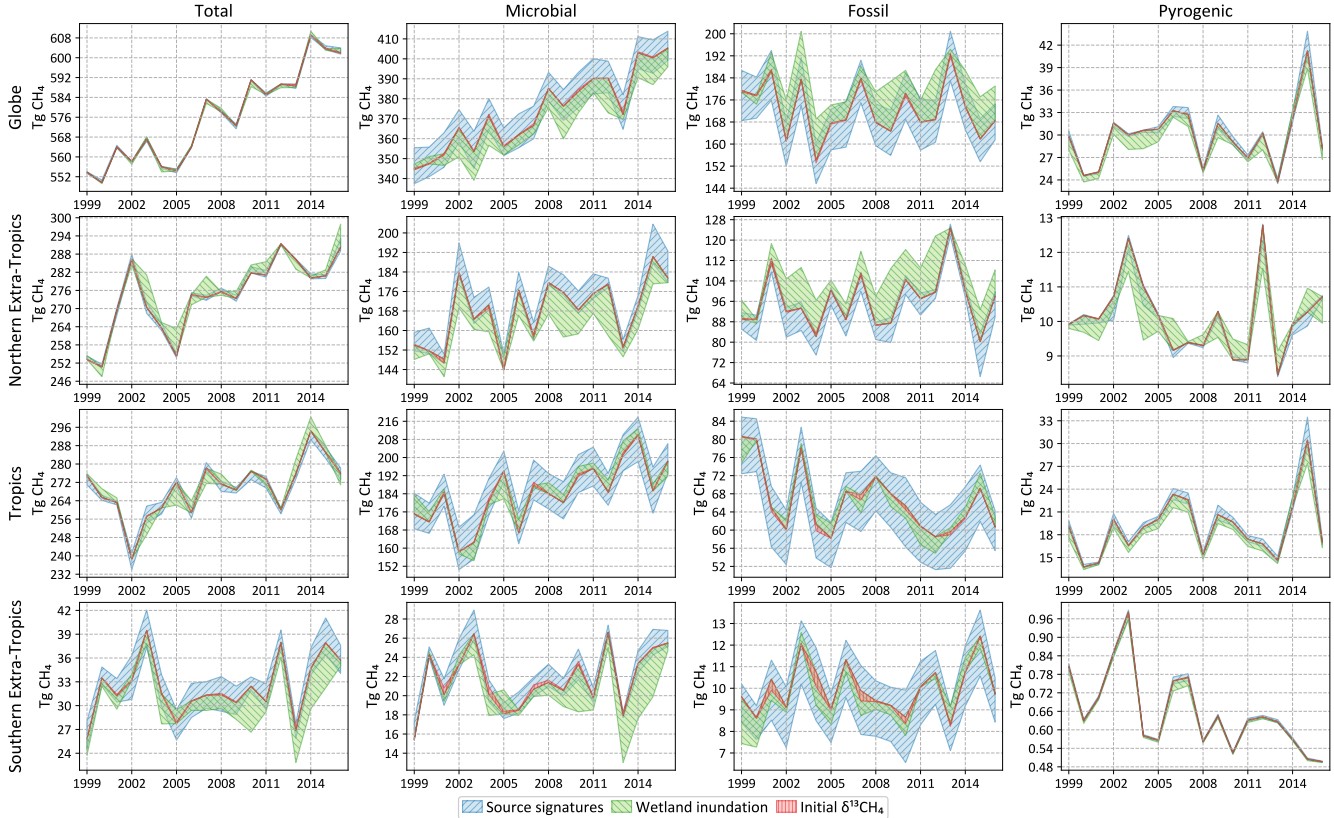

**Figure 9.** Total and source-specific annual emissions of $CH_4$ globally and from three latitudinal bands as in Figure 7. The shaded regions denote the spread (max to min) of annual emissions from sensitivity tests described in § 2.6.3, § 2.6.4 and § 2.6.5.

"incorrect" initial fields we constructed for the last test still satisfied the global $\delta^{13}CH_4$ mass balance by construction. The sensitivity to an incorrect initial condition will likely be higher if the initial field does not satisfy global $\delta^{13}CH_4$ mass balance.

### 3.4 Attribution of the post-2007 methane growth

As discussed earlier (Figure 1 and discussion in § 1), the atmospheric methane burden has been steadily growing since 2007 after a period of quasi-stability during 1999–2006. We use our $CH_4 + \delta^{13}CH_4$ inversion to ask whether the addition of $\delta^{13}CH_4$ data can provide information on the sources of the additional methane. Figure 7 suggests that the trend in $CH_4$ emissions comes largely from microbial emissions in a $CH_4 + \delta^{13}CH_4$ inversion. However, it is possible that this attribution to microbial emissions comes from our prior – which had a trend in the microbial emissions and a temporally flat fossil contribution –

instead of the atmospheric data. To assess the robustness of our inferred microbial and fossil emission trends, we perform a second set of inversions with the following modifications:



**Table 5.** Average annual Monte Carlo-derived random uncertainty and possible bias in $CH_4$ emissions, separated by source type (Tot = total, Mic = microbial, Fos = fossil, Pyr = pyrogenic) and latitude bands as in Figure 7. For each source type and region, the mechanism behind the largest possible bias has been demarcated by highlighting the bias in red. The "MC-derived" numbers are $2\sigma$ posterior uncertainties, all other numbers represent the range between maximum and minimum estimates. All numbers are in Tg $CH_4$ per year.

| **Region** | Globe | | | | N. Extra-Tropics | | | | Tropics | | | | S. Extra-Tropics | | | |
|---|---|---|---|---|---|---|---|---|---|---|---|---|---|---|---|---|
| **Source type** | Tot | Mic | Fos | Pyr | Tot | Mic | Fos | Pyr | Tot | Mic | Fos | Pyr | Tot | Mic | Fos | Pyr |
| MC-derived ($2\sigma$) | 3.8 | 13.1 | 13.5 | 2.9 | 9.5 | 13.7 | 12.9 | 1.5 | 10.7 | 12.9 | 10.4 | 2.3 | 4.3 | 5.0 | 2.9 | 0.2 |
| Tropospheric chlorine | 0.6 | 26.3 | 25.9 | 0.2 | 4.8 | 20.1 | 15.1 | 0.2 | 3.4 | 7.3 | 10.6 | 0.2 | 1.6 | 1.4 | 0.3 | 0.0 |
| OH fractionation | 1.5 | 42.1 | 41.4 | 0.8 | 1.7 | 10.5 | 11.9 | 0.1 | 3.0 | 29.2 | 26.1 | 0.6 | 1.3 | 2.5 | 3.4 | 0.0 |
| Source signatures | 1.1 | 16.3 | 16.2 | 1.0 | 1.8 | 8.8 | 8.2 | 0.2 | 4.1 | 16.3 | 12.3 | 1.2 | 3.5 | 2.1 | 2.6 | 0.0 |
| Wetland inundation | 1.0 | 9.3 | 10.1 | 1.3 | 3.5 | 8.9 | 10.6 | 0.6 | 3.7 | 4.6 | 2.3 | 1.1 | 2.5 | 2.1 | 0.7 | 0.0 |
| Initial $\delta^{13}CH_4$ | 0.1 | 0.5 | 0.4 | 0.0 | 0.1 | 0.4 | 0.4 | 0.0 | 0.1 | 0.4 | 0.4 | 0.0 | 0.1 | 0.2 | 0.1 | 0.0 |

1. We construct climatological prior fluxes and source signatures by averaging our prior emissions and signatures from 2000 to 2006. Neither the resulting priors nor the source signatures have any time trend.

2. Since the methane budget from climatological priors is no longer in balance with the atmospheric growth, we cannot use the overlapping inversions of Figure 2(a) to run multiple periods in parallel. Instead, we run four 6-year inversions in sequence, spanning 1997–2003, 2002–2008, 2007–2013 and 2012–2018, following the scheme shown in Figure 2(b). The first inversion used the same initial field in 1997 as our default inversion. Every successive inversion used the previous inversion's fifth year mole fraction field as initial condition. The last year of each inversion is discarded in the end, and the first five years' fluxes are stitched together and analyzed.

The posterior uncertainties of the emissions derived from this modified setup are calculated by performing a Monte-Carlo suite of 100 inversion as described in § 2.6. The Monte-Carlo runs follow the geometry of Figure 2(b) as well, with the i[th] inversion (i = 1 to 100) of each period initialized from the 5[th] year mole fraction field of the i[th] inversion of the previous period. This allows us to calculate not only annual uncertainties but also uncertainties on long term averages.

To study the transition around 2007, we considered two periods 2000–2006 and 2008–2014. Average total and source-specific emissions over the two periods are shown in Figure 10, as well as the change in the average emissions between the two periods. The prior fluxes do not change between the two periods, therefore the estimated change must be driven by the atmospheric observations. Both the $CH_4$-only and the $CH_4 + \delta^{13}CH_4$ inversions estimate a change in the total emission of $(27.1 \pm 0.6) \, \mathrm{Tg \, yr^{-1}}$ to match the increase in the atmospheric burden. However, while the $CH_4$-only inversion attributes $\sim 70\%$ of that to fossil $CH_4$ emissions and only $\sim 29\%$ to microbial emissions, the addition of $\delta^{13}CH_4$ data switches the balance to $\sim 15\%$ fossil and $\sim 85\%$ microbial. This change in the allocation of the methane emission in the presence of $\delta^{13}CH_4$ data is significant compared to the uncertainties on the changes as depicted in Figure 10. The contribution of pyrogenic emissions to





the change is small in both inversions, and its change between the two inversions is not significant compared to its uncertainty. This is consistent with the downward trend in the global average $\delta^{13}CH_4$ in Figure 1, since microbial sources are the lightest of the three source types.

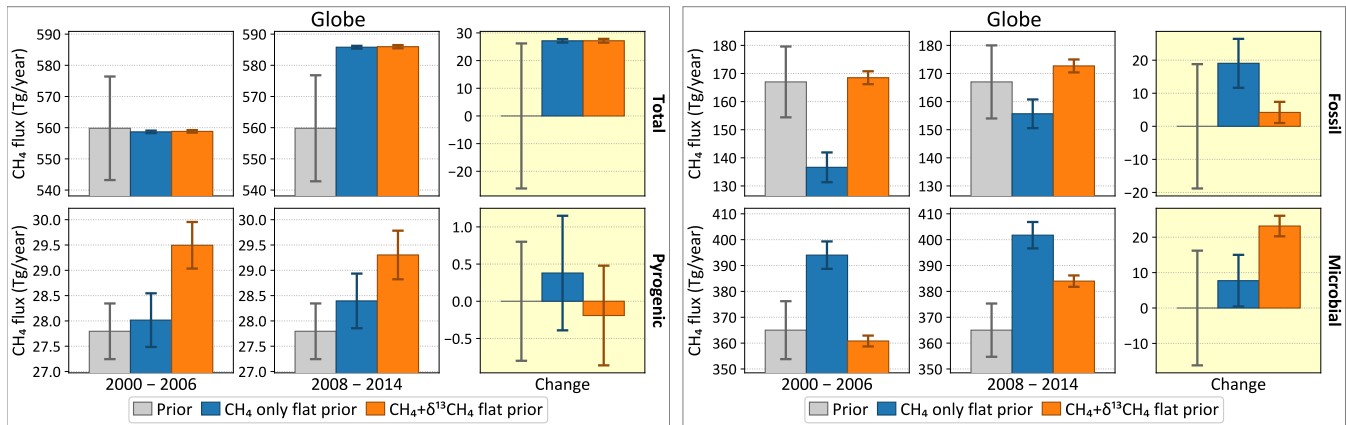

**Figure 10.** Change in global $CH_4$ emissions between the periods 2000–2006 and 2008–2014, total (top left) and disaggregated by source type. The gray bars denote prior emissions, and the colored bars denote two inversions, one with and the other without assimilated $\delta^{13}CH_4$ data. For each source type, the first two columns show the average emission over the two periods in question, and the third column shows the change between the two periods. The $1\sigma$ error bars are derived from a 100-member Monte Carlo ensemble of inversions following the configuration of Figure 2(b).

Geographically, the change between the two periods is driven almost equally by the Tropics and the northern extra-Tropics (Figure 11). In the Tropics, the addition of $\delta^{13}CH_4$ data results in higher microbial emissions in both periods. The change between the two periods is also attributed to microbial emissions, unlike a $CH_4$-only inversion which attributes the change primarily to fossil methane. In the northern extra-Tropics, although the presence of $\delta^{13}CH_4$ data point to increase in microbial emissions between the two periods, the relative apportionment of the increase between fossil and microbial emissions does not differ significantly from the $CH_4$-only emission if we consider the respective uncertainty estimates. This suggests that the global increase in microbial emissions between the two periods (Figure 10) is driven largely by the Tropics.

It is worth noting here that a change in emission strengths is not the only possible mechanism for an increase in atmospheric $CH_4$; a reduction in the sink strength could also induce a positive trend in atmospheric $CH_4$ post-2007. However, Lan et al. (2021) have shown that the changes in sinks proposed so far in the literature to explain the post-2007 $CH_4$ growth are not consistent with the observed $\delta^{13}CH_4$ trend post-2007. We therefore do not consider those alternatives here.

### 3.5 Separating microbial and fossil emissions

The $CH_4$ observations assimilated in a $CH_4$-only inversion constrain the total $CH_4$ emission, and any source disaggregation relies on spatiotemporal separation of emissions as encoded in the prior emissions and their uncertainties. Since the two largest $CH_4$ source types, microbial and fossil, have different $\delta^{13}CH_4$ source signatures, assimilating $\delta^{13}CH_4$ observations should





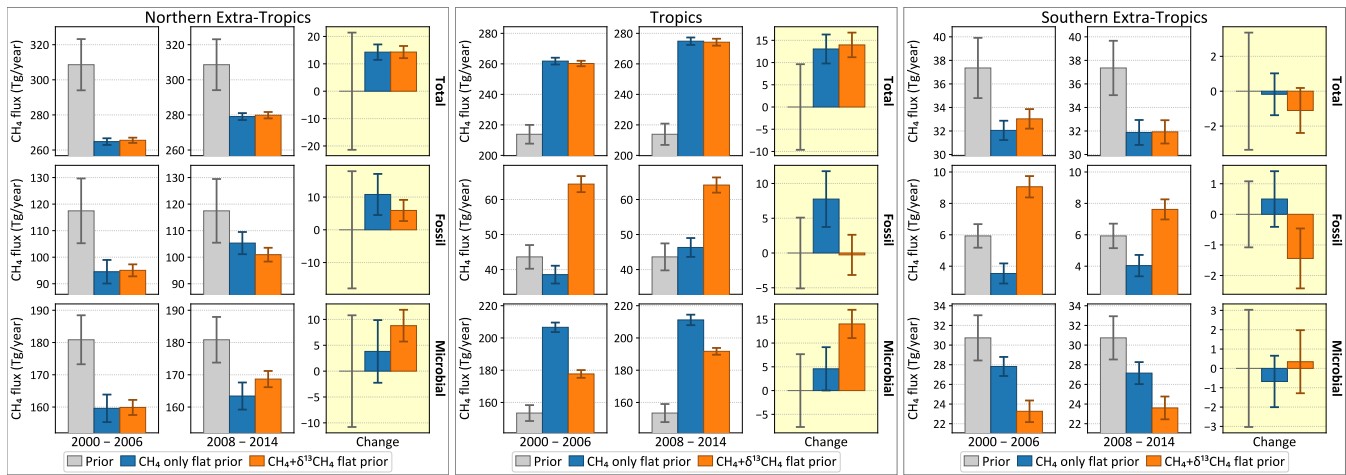

**Figure 11.** Similar to Figure 10, but disaggregated into latitude bands. As in Figure 7, "Tropics" refers to the region between 23.5 °S and 23.5 °N. Pyrogenic emissions have not been plotted because of their small contribution in all three latitude bands.

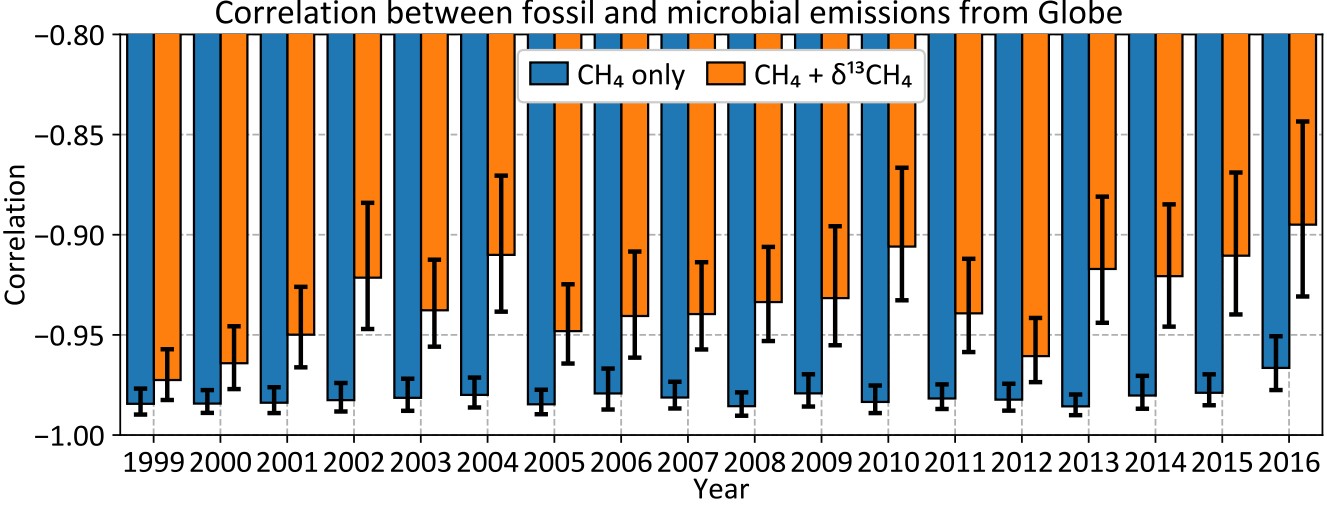

**Figure 12.** Posterior correlation between global annual microbial and fossil $CH_4$ emissions for the two inversions of Figure 7.

provide additional information to separate the two sources compared to a $CH_4$-only inversion. We can evaluate this additional information by looking at the posterior correlation between microbial and fossil emissions, both globally and regionally. Posterior correlations between global annual microbial and fossil $CH_4$ emissions, calculated from our 100-member ensemble of independent inversions as described in § 2.6, are shown in Figure 12. Error bars on the correlations shown in Figure 12 represent the 95th percentile range of 20,000 evaluations of the correlation by randomly sampling the 100-member inversion

ensemble with replacement (Efron and Tibshirani, 1994).





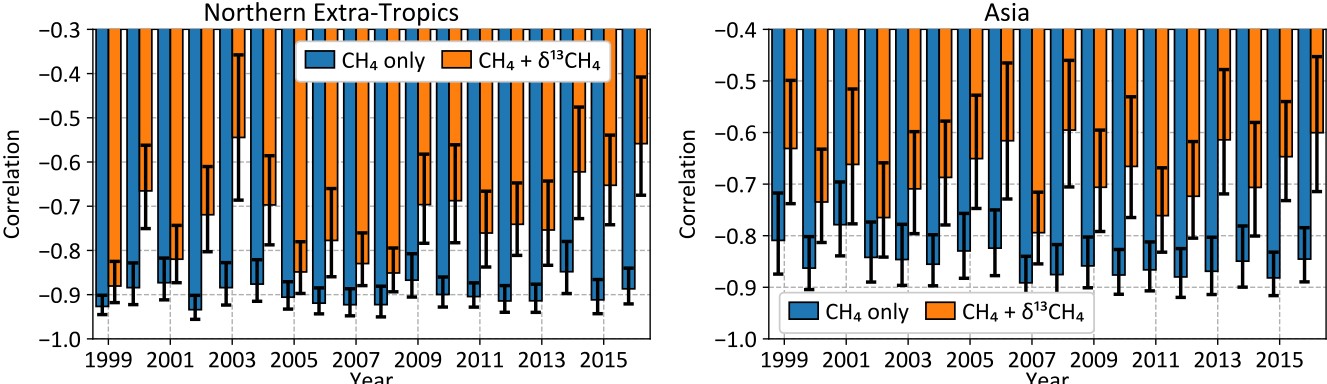

**Figure 13.** Posterior correlation between annual microbial and fossil $CH_4$ emissions over the Northern Extra-Tropics (left) and Asia (right) for the two inversions of Figure 7.

For all the years shown in Figure 12, a $CH_4$-only inversion results in a strong negative correlation between microbial and fossil emissions, consistent with the idea that atmospheric $CH_4$ measurements constrain the total $CH_4$ budget much better than individual source types. The addition of $\delta^{13}CH_4$ data reduces this negative correlation, implying that $\delta^{13}CH_4$ provides additional information to disentangle different $CH_4$ source types. The degree of disentanglement, represented by the reduction

in the negative correlation, is determined by the $\delta^{13}CH_4$ measurement coverage in a particular year and atmospheric transport connecting the emissions to those measurements. The correlation reduction is limited in our inversions by the relative sparsity of $\delta^{13}CH_4$ measurements; even in the most recent 2012–2017 inversion period, only 2.8 % of $CH_4$ measurements have corresponding $\delta^{13}CH_4$ measurements, overwhelmingly in locations far removed from significant $CH_4$ emissions (Figure 3). Having more $\delta^{13}CH_4$ measurements in general, and specifically closer to emissive regions, should allow further disentangling of the

different $CH_4$ source types.

Over smaller regions, only the Northern Extra-Tropics and Asia show significant decorrelation between annual fossil and microbial emissions (Figure 13) with the addition of $\delta^{13}CH_4$ data. While several other regions show similar reductions, the reductions are typically not significant compared to the 95[th] percentile error bars. The significant decorrelation seen for Northern Extra-Tropical and Asian emissions may be because most $\delta^{13}CH_4$ measurements are in the Northern Extra-Tropics and down-

wind of Asia in the Pacific. To see similar significant decorrelation over other regions we will likely need increased $\delta^{13}CH_4$ coverage closer to those regions. Although Figure 3 shows some $\delta^{13}CH_4$ measurements over North America and Europe, the majority of those measurements are from the background air sampling sites Niwot Ridge and Jungfraujoch respectively, and therefore do not contribute to significant decorrelation of fossil and microbial emissions from those continents.

### 3.6    Comparison to the GCP methane budget

The Global Carbon Project (GCP) periodically publishes top-down and bottom-up budgets of methane emissions from a suite of models. However, a meaningful comparison between our emissions and the 2020 GCP budget (Saunois et al., 2020) is



not straightforward. The GCP bottom-up (BU) budget for 2008–2017, with $737\,\mathrm{Tg\,yr^{-1}}$ emissions and $625\,\mathrm{Tg\,yr^{-1}}$ sinks, significantly overestimates the atmospheric growth rate. This is primarily due to an overestimate of both microbial ($159\,\mathrm{Tg\,yr^{-1}}$ freshwater sources) and fossil ($45\,\mathrm{Tg\,yr^{-1}}$ geologic sources) methane in the GCP budget, making a direct comparison with our

microbial and fossil estimates meaningless. The GCP top-down (TD) estimates do not provide a fossil-microbial split of "other natural" emissions, also making a direct comparison with our estimates difficult. However, it is possible to calculate emissions for certain GCP categories from our inversions for some limited comparisons.

Assuming that methane from geological seeps do not change significantly over decadal time scales, we subtract $35\,\mathrm{Tg\,yr^{-1}}$ geologic methane from our fossil methane emissions to estimate $(137\pm2)\,\mathrm{Tg\,yr^{-1}}$ fossil fuel emissions for both the 2000–

2009 and 2008–2016 periods, with a change of $(0\pm2)\,\mathrm{Tg\,yr^{-1}}$ between the two periods. This is in stark contrast with GCP BU estimates of $111\,\mathrm{Tg\,yr^{-1}}$ and $127\,\mathrm{Tg\,yr^{-1}}$ for the two periods respectively. The GCP TD fossil fuel estimates for the two periods are $99\,\mathrm{Tg\,yr^{-1}}$ and $109\,\mathrm{Tg\,yr^{-1}}$ respectively, lower than both our estimates and the GCP BU estimates. While our baseline estimates for the two periods may be influenced by systematic biases (§ 2.6), the change between the two periods is relatively robust. With the alternate specification of tropospheric chlorine (Wang et al., 2021) and alternate fractionation due to

the OH oxidation (Cantrell et al., 1990), the two biggest sources of bias in source apportionment by $\delta^{13}\mathrm{CH_4}$, the change in our fossil fuel emission estimate between the two periods is $1\,\mathrm{Tg\,yr^{-1}}$ and $-1.7\,\mathrm{Tg\,yr^{-1}}$ respectively, well within our uncertainty estimate of $2\,\mathrm{Tg\,yr^{-1}}$ and significantly lower than both the GCP BU and the GCP TD estimates. The GCP BU (TD) estimate of an increase of $16\,\mathrm{Tg\,yr^{-1}}$ ($10\,\mathrm{Tg\,yr^{-1}}$) between the two periods is closer to our estimate of $(8.4\pm5.6)\,\mathrm{Tg\,yr^{-1}}$ from a $\mathrm{CH_4}$-only inversion, but not when $\delta^{13}\mathrm{CH_4}$ data are incorporated.

For reasons mentioned above, we cannot directly compare our microbial emission estimates to GCP emission estimates. However, if we assume that methane from termites, wild animals and oceans do not change over decadal time scales, we can compare the change in the GCP TD estimate of wetlands, agriculture and waste from 2000–2009 to 2008–2016 with the change of microbial emissions in our inversion estimates. The GCP TD budget estimates a change of $12.6\,\mathrm{Tg\,yr^{-1}}$ between those two periods, compared to our estimate of $(26\pm2)\,\mathrm{Tg\,yr^{-1}}$ from a joint $\delta^{13}\mathrm{CH_4}$ and $\mathrm{CH_4}$ inversion and of $(18\pm6)\,\mathrm{Tg\,yr^{-1}}$ from a

$\mathrm{CH_4}$-only inversion. Thus, the change in microbial emissions in the GCP TD budget is at the lower end of but consistent with our estimate from a $\mathrm{CH_4}$-only emission, while it is not consistent with our budget after incorporating $\delta^{13}\mathrm{CH_4}$ data. We cannot perform a similar analysis with the GCP BU budget because freshwater emissions cannot be assumed to be static over decadal time scales.

Finally, our pyrogenic emission estimates for both 2000–2009 and 2008–2016 periods are $(30.0\pm0.6)\,\mathrm{Tg\,yr^{-1}}$, with a

change of $(0.3\pm0.5)\,\mathrm{Tg\,yr^{-1}}$. These are close to the GCP BU (TD) estimates of $31\,\mathrm{Tg\,yr^{-1}}$ and $30\,\mathrm{Tg\,yr^{-1}}$ ($29\,\mathrm{Tg\,yr^{-1}}$ and $31\,\mathrm{Tg\,yr^{-1}}$) respectively. Neither the GCP budgets nor our inversion show significant changes in pyrogenic methane emissions between the two periods.





## 4 Discussion

We have constructed a variational atmospheric inversion system capable of assimilating $CH_4$ and $\delta^{13}CH_4$ measurements to

estimate source-specific methane emissions within the TM5 4DVAR framework. We have assimilated $CH_4$ and $\delta^{13}CH_4$ measurements from a multi-agency air sampling network in this framework to estimate fossil, microbial and pyrogenic emissions of atmospheric $CH_4$ globally. We have derived Bayesian uncertainty estimates on our emissions (random error), as well as investigated the impact of biases from non-optimized aspects of our inversion (systematic error). Our conclusions can be summarized as follows.

First, Figures 5 and 6 show that our inversion assimilating only $CH_4$ does not yield a $CH_4$ emission distribution consistent with atmospheric $\delta^{13}CH_4$. This is very likely true of $CH_4$ inversions in general, since they have no constraints forcing them to match atmospheric $\delta^{13}CH_4$ gradients and trends. Starting from a prior emission distribution consistent with atmospheric $\delta^{13}CH_4$ trends does not ensure that the posterior emission estimates will remain consistent. Our $CH_4$-only inversion started from a prior that reproduced the global mean atmospheric $\delta^{13}CH_4$ trend (scenario "C_WL+" of Lan et al., 2021), yet the

posterior deviated from it as shown in Figures 5 and 6. We conclude that the only way to guarantee a posterior emission distribution consistent with both atmospheric $CH_4$ and $\delta^{13}CH_4$ data is to assimilate them simultaneously.

Second, given an atmospheric sink scenario, our current observational coverage allows us to estimate the global total $CH_4$ emission with a $2\sigma$ random uncertainty of $\sim 3.8\,\mathrm{Tg\,yr^{-1}}$, which is less than $1\,\%$ of the total emission. Microbial, fossil and pyrogenic emission uncertainties are around $3.5\,\%$, $8\,\%$ and $10\,\%$ respectively at the global scale. Given these posterior uncer-

tainties, there are significant differences between inversions with and without $\delta^{13}CH_4$ data in the apportionment of the total $CH_4$ emission between microbial and fossil sources, both globally and in the Tropics (Figure 7 and Table 4). In both regions, the inclusion of $\delta^{13}CH_4$ data in an inversion results in a significantly higher proportion of fossil emissions compared to microbial emissions, which we consider realistic since it matches both atmospheric $CH_4$ and $\delta^{13}CH_4$ data (Figures 4 and 5). Pyrogenic emissions are relatively insensitive to the inclusion of $\delta^{13}CH_4$ data.

Third, we tested the sensitivity of our results to several factors that can lead to biases or systematic errors, as detailed in § 2.6. This included different maps of the $\delta^{13}CH_4$ isotopic source signatures, static and dynamic maps of the wetland inundation extent, different initial $\delta^{13}CH_4$ fields, different fractionation factors for the $CH_4 + OH$ oxidation mechanism, and different fields of tropospheric Cl. The last two factors had by far the largest impacts on the large scale apportionment between microbial and fossil emissions, even though their impact on the total $CH_4$ budget was nil or negligible. With the OH fractionation of Cantrell

et al. (1990), the global microbial emission increases to $414\,\mathrm{Tg\,yr^{-1}}$ and the fossil emission drops to $131\,\mathrm{Tg\,yr^{-1}}$. With the lower estimate of tropospheric Cl from Wang et al. (2021), the global microbial emission decreases to $345\,\mathrm{Tg\,yr^{-1}}$ while the fossil emission increases to $199\,\mathrm{Tg\,yr^{-1}}$. Since some $CH_4$ inversions in the literature do not simulate a tropospheric Cl sink of $CH_4$, we tested the impact of this limiting case as well. In the absence of a tropospheric Cl sink, the global microbial emission drops further to $331\,\mathrm{Tg\,yr^{-1}}$ and the fossil emission increases to $213\,\mathrm{Tg\,yr^{-1}}$. Most of these shifts in the global

partitioning are accompanied by shifts in the latitudinal partitioning. All of these are significant revisions to the partitioning of Table 4, suggesting that the ability of atmospheric $\delta^{13}CH_4$ measurements to partition the total $CH_4$ emission into different





source types, at least over large regions, is limited by our knowledge of these two critical chemical processes. The uncertainty in our knowledge of $\delta^{13}CH_4$ source signatures, long considered a limitation on the use of $\delta^{13}CH_4$ data, is almost never a leading driver of uncertainty in Table 5, although it is usually more significant than either inundation extent or the initial $\delta^{13}CH_4$ field.

Finally, our tests suggest that the impact of an incorrect initial $\delta^{13}CH_4$ field can be ameliorated by a relatively short spin-up of one year in an inversion, in contrast to a multi-decadal spin-up necessary for a forward model run.

Fourth, atmospheric $\delta^{13}CH_4$ data strongly suggest that the rise in microbial emissions is the primary driver of the post-2007 growth in atmospheric $CH_4$. While a $CH_4$-only inversion starting from priors without a time trend attributes $\sim70\,\%$ of the growth to fossil emissions, the addition of $\delta^{13}CH_4$ data shifts that to microbial emissions being responsible for $\sim85\,\%$ of the

growth. Since the latter inversion is consistent with atmospheric $\delta^{13}CH_4$ data while the former is not (Figure 5), we consider a majority microbial contribution to the post-2007 growth to be more realistic. A disaggregation of the growth by latitude bands suggests that a significant majority of the increase in tropical methane emissions is due to microbial and not fossil emissions. Moreover, although some of the sensitivity tests of § 2.6 lead to different partitioning between microbial and fossil emissions, they all suggest a steeper trend in microbial compared to fossil emissions in Figures 8 and 9.

Fifth, the ability of $\delta^{13}CH_4$ data to disentangle different $CH_4$ source types can be quantified by the reduction in the posterior correlation between emissions from those sources owing to the addition of $\delta^{13}CH_4$ data, compared to a $CH_4$-only inversion. Considering the two largest source types of methane, microbial and fossil, we see significant reductions in their posterior correlation over the globe as well as the northern extra-tropics and Asia. The degree of decorrelation, however, is limited, and we do not see significant decorrelation over other regions. We hypothesize that this is not a limitation of our understanding

of $\delta^{13}CH_4$ but rather of its limited observational coverage. Even in the most recent years less than $3\,\%$ of assimilated $CH_4$ measurements were accompanied by $\delta^{13}CH_4$ measurements, almost exclusively from background sites. It is very likely that an increase in the observational coverage of $\delta^{13}CH_4$, preferably close to source regions, will improve the capability of $\delta^{13}CH_4$ measurements to distinguish between different $CH_4$ source types.

Sixth, while it is difficult to compare our emission budget directly with GCP due to different partitioning schemes, we note

that our fossil fuel emissions for both the 2000–2009 and 2008–2016 periods are higher than the GCP top-down and bottom-up emissions. However, our estimate of the change in fossil fuel emissions between the two periods is significantly lower than the GCP estimates. Concurrently, our estimate of the change in microbial emissions over the same time is significantly higher than the GCP top-down estimate. Both of these discrepancies are driven by atmospheric $\delta^{13}CH_4$ data, since our $CH_4$-only inversion provides changes that are consistent with GCP estimates. We therefore conclude that the microbial and fossil

emission change estimates in the GCP budget are consistent with atmospheric $CH_4$ data but not with $\delta^{13}CH_4$ data. Finally, our pyrogenic emission estimates are consistent with or close to the GCP estimates for both periods.

## 4.1 Comparison with Thanwerdas et al. (2021)

Thanwerdas et al. (2021) describe an alternative variational inversion framework using the LMDz-SACS model to assimilate $CH_4$ and $\delta^{13}CH_4$ measurements. We find it heartening that others have decided to tackle this complicated problem. Since they

reserve decadal dual tracer inversions for future work, we will compare their technique with ours to highlight the similarities





and differences. The biggest difference lies in the decision of Thanwerdas et al. (2021) to optimize $\delta^{13}CH_4$ source signatures, compared to our choice of keeping them fixed for a specific inversion. While $\delta^{13}CH_4$ source signatures are uncertain for many methane sources, we explain our reasons for not optimizing them in § 2.6.3. Instead, we explore the impact of source signature uncertainty with different constructions of the source signature map as detailed in § 2.6.3. In the end, at least for large

geographical regions, the uncertainty from source signatures did not prove to be a leading uncertainty (Table 5). The second major difference between the two inversion frameworks lies in the construction of the prior $CH_4$ fluxes. While Thanwerdas et al. (2021) use a prior that approximately matches the atmospheric $CH_4$ growth rate, we construct our priors to match both the $CH_4$ growth rate and the $\delta^{13}CH_4$ trend over two decades. We suspect this, and the linearity of our formulation due to not optimizing source signatures, to be the reasons why our inversion required a shorter spin-up time compared to Thanwerdas

et al. (2021).

There are also a few differences in implementation between the two frameworks. Most notably, Thanwerdas et al. (2021) estimate the posterior uncertainty as the spread between different inversion configurations, correctly stating that an evaluation of the posterior covariance matrix would require significantly more computing resources. We evaluate that posterior covariance matrix for both $CH_4 + \delta^{13}CH_4$ and $CH_4$-only inversions and present both types of uncertainty, namely the systematic uncertainty

as the spread between multiple inversion configurations, and the random (Bayesian) uncertainty as the spread of an ensemble of 100 independent inversions. The configurations we explore for the systematic uncertainty are also different from Thanwerdas et al. (2021), and include alternate specifications of the Cl oxidant and the isotopic discrimination of the $CH_4 + OH$ reaction. We find the latter two to be the most significant drivers of uncertainty for partitioning $CH_4$ emissions using $\delta^{13}CH_4$ data.

While our implementation of the inversion is different from Thanwerdas et al. (2021), our goals are very similar. We look

forward to long-term inversions of $CH_4$ and $\delta^{13}CH_4$ data using LMDz-SACS so that we may compare and contrast with our results presented here, and figure out how best to use isotopic measurements to solve the atmospheric methane puzzle.

## 4.2 Future work and outlook

While we feel confident in the $CH_4$ emission estimates reported here, there are several areas which we plan to explore and improve in future work.

### 4.2.1 Alternate OH

The atmospheric $CH_4$ budget is determined by the balance between its sources and sinks, the latter primarily driven by the OH radical. While there have been some efforts to optimize atmospheric OH in concert with $CH_4$ emissions (e.g., Zhang et al., 2018, 2021; Yin et al., 2021), we do not think in situ $CH_4$ samples provide sufficient information to constrain the sink independently. Moreover, estimates of OH abundance and variability over the past decades, either from $CH_4$ inversions (Yin

et al., 2021) or otherwise (Bousquet et al., 2005; Montzka et al., 2011; Nicely et al., 2018), are consistent with a limited role of OH variability in recent trends in atmospheric $CH_4$. This is why, similar to the vast majority of $CH_4$ inversions, we have chosen to keep the OH sink fixed to a field consistent with observed trends and gradients of methyl chloroform (MCF, Spivakovsky et al., 2000; Patra et al., 2014, 2020). Nonetheless, we acknowledge that our knowledge of atmospheric OH is imperfect and





uncertain, and in future work we plan to explore alternate specifications of OH that are consistent with our knowledge of atmospheric chemistry and MCF trends and gradients.

### 4.2.2 Alternate optimizer and source signature uncertainty

Errors in the specification of the $\delta^{13}CH_4$ source signatures can have significant impact on the inferred methane emissions (Thanwerdas et al., 2021). While we have explored alternate specifications, it is possible that the true uncertainty in $\delta^{13}CH_4$ source signatures is larger than the range we have explored. Optimizing the $\delta^{13}CH_4$ source signatures with a realistic prior covariance structure may yield larger but more realistic error bounds on source-specific methane emissions. We plan to explore that option in the future, which will require an alternate to the conjugate gradient optimizer (Lanczos, 1950) we currently use. We have tested the M1QN3 optimizer used by Thanwerdas et al. (2021), and have found its convergence to be slow and inefficient for our system. Therefore, we plan to explore and implement alternate optimizers that can work efficiently on non-linear problems, in order to have the option of estimating $\delta^{13}CH_4$ source signatures. Concurrently, we will work on a more complete characterization of the $\delta^{13}CH_4$ source signature uncertainty, which will be required in order to derive a prior error covariance matrix for $\delta^{13}CH_4$.

### 4.2.3 OSSEs

We have tested the ability of existing $\delta^{13}CH_4$ observations to infer mechanisms behind the recent $CH_4$ growth and separate different $CH_4$ source types, and found that the ability to distinguish fossil from microbial emissions – as reflected by the posterior correlation between them – is limited at policy-relevant scales (§ 3.5). We strongly suspect that this is a limitation of the existing $\delta^{13}CH_4$ observational coverage and not of the inversion technique. If we consider expanding the $\delta^{13}CH_4$ measurement network to improve that ability in the future, we need to quantify the added value of different expansion strategies. We plan to do this with Observation System Simulation Experiments (OSSEs) simulating different observational networks, as we have done for $^{14}C$ of $CO_2$ in the past (Basu et al., 2016).

### 4.2.4 Satellite $CH_4$ retrievals

Several satellites have been launched by various space agencies in the past decades to estimate atmospheric $CH_4$ from space, and several more are slated to go up over the next decade. As the technique to use $\delta^{13}CH_4$ in $CH_4$ inversions matures, we hope to eventually add satellite $CH_4$ data to such inversions to provide stronger regional constraints.

*Code availability.* TM5 4DVAR code for performing the inversions is publicly available at https://sourceforge.net/p/tm5/cy3_4dvar/ci/ default/tree/.

*Data availability.* The $CH_4$ and $\delta^{13}CH_4$ data assimilated for this exercise can be downloaded from https://doi.org/10.15138/64w0-0g71.





*Author contributions.* SB built the extensions needed to TM5 4DVAR to assimilate $CH_4$ and $\delta^{13}CH_4$ data, and performed all model runs. SS and XL constructed prior flux and source signature maps. ED and SM provided $CH_4$ and $\delta^{13}CH_4$ data respectively, with expert advice on data errors and which datasets to assimilate. ED, SM, SS and XL constructed the multi-agency atmospheric data sets to be assimilated. KT collated all measurements into a common format for model use. SS obtained the initial grant for this study and designed the study protocols with the other co-authors. PPT and JBM provided expertise on interpreting $\delta^{13}CH_4$ measurements and the formulation of the isotope mass balance equations. LB provided expertise on the global methane budget. YO provided wetland $CH_4$ emissions and source signatures from the IsoTEM model. FA provided $CH_4$ data from Plateau de Rosa, Italy. LVG provided $CH_4$ data from multiple aircraft profiling sites in Brazil. AJ provided $CH_4$ and $\delta^{13}CH_4$ data from several sites maintained by the Max Planck Institute for Biogeochemistry, Jena. JN provided $CH_4$ data from Kasprowy Wierch, Poland. MS provided $CH_4$ and $\delta^{13}CH_4$ data from multiple surface sites, shipboard and aircraft sampling programs run by the National Institute for Environmental Studies, Japan. SM provided $CH_4$ and $\delta^{13}CH_4$ data from surface and shipboard sampling programs run by Tohoku University. TDI provided $CH_4$ data from several sampling sites in Italy. GM provided $CH_4$ data from Ispra, Italy. HL provided $CH_4$ data from Anmyeon-do, Korea. JA provided $CH_4$ data from Monte Cimone, Italy. The manuscript was primarily written by SB, with input from XL, ED, SM, SS, JBM and PPT.

*Competing interests.* The authors declare no competing interests.

*Acknowledgements.* This work was supported by funding from the National Aeronautics and Space Administration (NASA), grant NNX17AK20G. All computing work was performed on either NASA's Discover supercomputer at the NASA Center for Climate Simulation (NCCS) or NOAA's Orion supercomputer maintained by the Mississippi State University. SB was additionally supported by NASA's Modeling, Analysis and Prediction and Carbon Monitoring System Programs. XL was supported in part by cooperative agreement NA17OAR4320101 between NOAA and the University of Colorado at Boulder. In addition to the coauthors who provided atmospheric $CH_4$ and $\delta^{13}CH_4$ measurements, the authors acknowledge the work of (i) Ove Hermansen, Cathrine Lund Myhre, and Stephen Platt of the Norwegian Institute for Air Research (NILU) in collecting air samples at the Zeppelin Observatory through Norwegian Environment Agency grant 21087006, ICOS Norway, Research Council of Norway grant 296012, and ReGAME, Research Council of Norway grant 325610; (ii) Heiko Moosen and Willi A. Brand of the Max Planck Institute for Biogeochemistry, funded by the Max Planck Society; (iii) Doug Worthy of Environment Climate Change Canada; (iv) Casper Labuschagne of the South African Weather Service; (iv) László Haszpra of the Hungarian Meteorological Service; (v) Yosuke Niwa and Taku Umezawa of the National Institute for Environmental Studies, Japan; (vi) Shinya Takatsuji of the Japan Meteorological Agency for $CH_4$ measurements at Ryori, Yonagunijima, Minamitorishima and from JMA's "Aircraft Observation of Atmospheric trace gases" program; (vii) Arlyn Andrews, Bianca Baier, Molly Crotwell, Philip Handley, Jack Higgs, Jon Kofler, Pat Lang, Thomas Legard, Kathryn McKain, Eric Moglia, Don Neff, Tim Newberger, Colm Sweeney, and Sonja Wolter for support of the NOAA GGGRN tower and aircraft programs; (viii) Nina Paramonova of the Voeikov Main Geophysical Observatory, Russia; (ix) Dagmar Kubistin of the Deutscher Wetterdienst (DWD); (x) Karin Uhse and Ludwig Ries of the Umwelt Bundesamt, Germany; (xi) Juha Hatakka of the Finnish Meterological Institute; (xii) Emilio Cuevas of the Meteorological State Agency of Spain (AEMET); (xiii) Alex Vermeulen of Lund University; and (xiv) John Moncrieff of the University of Edinburgh. The authors also acknowledge atmospheric data provided by the National Institute of Water and Atmospheric Research (NIWA) of New Zealand, the Commonwealth Scientific and Industrial Research Organisation (CSIRO) of Australia, Laboratoire des Sciences du Climat et de l'Environnement (LSCE) of France, and the Swiss Federal





Laboratories for Materials Science and Technology (EMPA). We also thank Martin Steinbacher (EMPA) for comments on the manuscript. NIWA measurements at Arrival Heights and Baring Head were supported by internal funding under Climate and Atmosphere Research Programme CAAC2204 (2021/22 SCI). EMPA measurements at Jungfraujoch were supported by the Swiss National Air Pollution Monitoring Network, the Federal Office for the Environment and ICOS Switzerland (Swiss National Science Foundation, grant 20FI21_148992). RSE

contribution to this work has been financed by the Research Fund for the Italian Electrical System under the Contract Agreement between RSE S.p.A. and the Ministry of Economic Development – General Directorate for the Electricity Market, Renewable Energy and Energy Efficiency, Nuclear Energy in compliance with the Decree of April 16th, 2018.



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
