# Peer review of "Estimating Emissions of Methane Consistent with Atmospheric Measurements of Methane and $\delta^{13}$ C of Methane"

_Atmospheric Chemistry and Physics, 2022_

## Author Response (AR1)

Dear Editor,

We have uploaded point-by-point responses to all three reviewers in the public discussion section of the manuscript. The reviews of Reviewer #2 (R2) and Dr. Manning were overwhelmingly positive, and we are glad that they consider our effort a progress in the field. Reviewer #1 (R1) recommended major revisions; however, in going through their comments, we did not really see that significant revisions were necessary – in the form of additional model runs or additional analysis – to respond to the comments and concerns. As far as we could understand, R1 had two major concerns, (1) our construction of the prior and prior uncertainty, and (2) the role of OH in explaining the post-2007 $CH_4$ growth.

**Construction of prior:** The reviewer is concerned by our approach of balancing the prior to follow the atmospheric growth rate of $CH_4$. This is standard practice for long-lived greenhouse gases, especially for long inversions where an unbalanced prior can lead farther and farther from the truth (Houweling et al., 2017; Chevallier et al., 2010; Weir et al., 2021; Patra et al., 2011). The concern about using the data twice in an inversion is (in our opinion) a red herring that may be true in theory but irrelevant in practice. As we have illustrated in Figure 1 (also included in response to R1), we could have balanced the prior using just the growth rate inferred from $CH_4$ flask samples at Mauna Loa (**not** assimilated in our inversion, red squares) and obtained almost identical prior flux totals as we currently have from using the marine boundary layer (MBL) growth rate (blue circles). To run the numbers, let us pick the *maximum* difference between the red and blue points in any year in Figure 1, ~4 ppb/year, which translates to ~10 Tg $CH_4$/year. So if we had used the Mauna Loa-based growth rate to construct our prior, it would have been different from our current prior by *not more than* 10 Tg/year. According to Table 4 in our manuscript, the 1σ prior uncertainty on our global total is 40 Tg/year, significantly larger than this difference. Also according to Table 4, the posterior 1σ uncertainty is 2 Tg/year, denoting an uncertainty reduction of 95%. Given this tight constraint on the global growth rate imposed by the data, and the generous prior uncertainty (compared to the 10 Tg/year potential difference between Mauna Loa-based and MBL-based growth rates), we cannot imagine how our results would have been any different had we used a Mauna Loa-based growth estimate to balance prior fluxes instead of the MBL-based estimate.

R1 also raises the concern that our 1σ prior uncertainty of 40 Tg/year is smaller than the gap of 160 Tg/year between the top-down and bottom-up estimates of the Global Carbon Project (GCP; Saunois et al (2020)). This is, frankly, a confusing comment since the two numbers are not supposed to be related in a Bayesian inversion. The 160 Tg/year gap reported by Saunois et al (2020) is a *bias*. The prior uncertainty in a Bayesian inversion is not supposed to reflect the size of a bias, it is supposed to reflect the covariance of (prior – truth), assuming that the prior, or more accurately, the ensemble of all priors, is *unbiased*. This is also why we have used the atmospheric growth rate to balance the prior, since otherwise we would be starting from a biased prior. And finally, we note that a 1σ uncertainty of 40 Tg/year implies that 95% of all possible priors lie within a range of 160 Tg/year. So *even if* we wanted to specify a prior uncertainty to encompass the GCP-reported bias – which is not required mathematically – 40 Tg/year (1σ) is not unreasonably small.

**OH variability:** We are aware that variations in atmospheric methane can be caused either by changes in the source or the sink, and separating

[Figure]

**Figure 1** Annual atmospheric $CH_4$ growth rate calculated from (blue) marine boundary layer samples and (red) daytime flask samples collected at Mauna Loa, Hawaii. The red dashed lines represent five-year average growth rates from either.

the two has been a long standing problem in methane studies. Several previous top-down studies have attributed part or all of the post-2006 $CH_4$ growth to a negative trend in atmospheric OH (Rigby et al., 2017; Turner et al., 2017; Worden et al., 2017; McNorton et al., 2018). However, *none* of those studies have presented a mechanistic understanding or hypothesis on *why* OH should have a sustained negative trend over 15 years. In fact, studies based on our best knowledge of atmospheric chemistry and recorded meteorology over those 15 years have concluded that such a sustained trend is highly unlikely (Nicely et al., 2018; Anderson et al., 2021).

Another point to note here is that a $CH_4$ growth based on a reduction of OH cannot explain the downward trend in atmospheric OH. The relationship between the emission-weighted source signature $\delta_s$ and the atmospheric isotope ratio $\delta_a$ at steady state (when sources and sinks are balanced) is $\delta_s = \alpha\delta_a - \varepsilon$, where $\alpha = k_{12}/k_{13} > 1$ is the ratio of oxidation rates of $^{12}CH_4$ and $^{13}CH_4$, and $\varepsilon = 1000\times(\alpha-1) > 0$ (Miller, 2004). Notably, this relationship does not depend on the total sink magnitude. So if OH were the only sink of $CH_4$, a reduction in OH would not explain a downward trend in $\delta^{13}CH_4$. In fact, OH is not the only sink of $CH_4$. While is the largest sink, it is also the sink that discriminates the least (has the smallest $\alpha$). Therefore, a reduction in OH without changing the other sinks would *increase* the sink-weighted $\alpha$, which would also *increase* $\varepsilon$, and thereby *increase* $\delta_a$ given the same sources. We have already demonstrated this in an earlier publication (Lan et al., 2021). We do not claim that OH variations cannot have a role in the interannual variability of $CH_4$, we know it must. We are only saying, after careful consideration and based on atmospheric chemistry studies and isotope mass balance, that a negative trend in OH is highly unlikely to be an important driver behind the post-2007 $CH_4$ growth. Therefore, an assessment of the current work that we are taking a "too easy" way out is, in our opinion, a criticism that is too easy to level and one that ignores the difficulties with an OH-driven explanation for 15+ years of methane growth.

We have attached a "difference" PDF between our original manuscript and the revised manuscript, where deletions are struck out (underlined in math mode) in red and additions are underlined in blue. However, tracking differences is not without flaws, so in case the additions and changes are not clear please refer to the revised manuscript. Here we list the most significant changes in order in which they appear in the manuscript. Line numbers, where given, refer to line numbers in the revised manuscript.

1. Equation (1): R1 expressed concern that calling L the "inverse lifetime" might confuse some readers into thinking that it was the lifetime, when in fact it was the loss rate. We have reworded the description to say "loss rate or inverse lifetime".
2. Mathematical fonts have been changed in equations (1), (2) and (3) to make them consistent throughout the text. The contents have *not* been changed.
3. Line 106: In response to R2, we have described $x$ as "the set of all $F_s$ of §2.1".
4. Line 142: In response to R2, we have fixed the citation. Wild animal emissions are taken from Houweling et al (1999) and termite emissions are from Sanderson (1996).
5. R2 expressed interest in knowing the numbers for a different definition of the Tropics often used in the methane literature, namely the band between 30°S and 30°N. We did not change the nomenclature in the manuscript at this stage since that might have introduced internal inconsistencies. However, we understand R2's motivation. We have therefore included emission estimates for three additional latitude bands (south of 30°S, 30°S to 30°N, north of 30°N) in Table 4, and included a "change in emissions" panel for 30°S-30°N in Figure 11, which we think are the two key places where an alternate definition of the Tropics becomes relevant.
6. Table 4 is supposed to summarize the numbers behind Figure 7. However, in the original manuscript, we made a few mistakes copying numbers by hand from our model runs to the table. We have fixed those mistakes. The numbers in Table 4 are now consistent with what is displayed in Figure 7.

7. Figure 8: Following recommendation from R1, we have added an explanation in the figure caption for why a smaller chlorine sink and a stronger OH fractionation result in opposing shifts in the fossil vs microbial split.
8. Line 406: Following the recommendation of R1, we have replaced "suggests" with stronger wording.
9. Figures 12 and 13 in the original manuscript were very similar. They have now been combined into a three-panel figure 12.
10. "Comparison to the GCP methane budget" has been removed from the Results section to §4.2.
11. All three reviewers wanted a reworking of the conclusions and discussion section. Accordingly, §4 is now called "Conclusions and discussion", with an enumerated list of conclusions from our work in §4.1. Section 4.2 now presents a comparison of our work to the GCP methane budget, while sections 4.3 to 4.5 present comparisons between our work and some other recent top-down studies involving $\delta^{13}CH_4$ data. Finally, in response to R2, we have created §4.7 with a summary of future needs for better utilizing atmospheric $\delta^{13}CH_4$ measurements that go beyond our future plans.

The above is a summary of the most important changes to our manuscript. Please refer to the (separately uploaded) public responses to the reviewers for point-by-point responses to their comments.

We hope that we have been able to address all reviewer concerns to your satisfaction. We look forward to the publication of our work in Atmospheric Chemistry and Physics in the near future.

Regards,

Sourish Basu and co-authors

References

Anderson, D. C., Duncan, B. N., Fiore, A. M., Baublitz, C. B., Follette-Cook, M. B., Nicely, J. M., and Wolfe, G. M.: Spatial and temporal variability in the hydroxyl (OH) radical: understanding the role of large-scale climate features and their influence on OH through its dynamical and photochemical drivers, Atmospheric Chem. Phys., 21, 6481–6508, https://doi.org/10.5194/acp-21-6481-2021, 2021.

Chevallier, F., Ciais, P., Conway, T. J., Aalto, T., Anderson, B. E., Bousquet, P., Brunke, E. G., Ciattaglia, L., Esaki, Y., Fröhlich, M., Gomez, A., Gomez-Pelaez, A. J., Haszpra, L., Krummel, P. B., Langenfelds, R. L., Leuenberger, M., Machida, T., Maignan, F., Matsueda, H., Morguí, J. A., Mukai, H., Nakazawa, T., Peylin, P., Ramonet, M., Rivier, L., Sawa, Y., Schmidt, M., Steele, L. P., Vay, S. A., Vermeulen, A. T., Wofsy, S., and Worthy, D.: $CO_2$ surface fluxes at grid point scale estimated from a global 21 year reanalysis of atmospheric measurements, J Geophys Res, 115, D21307–D21307, 2010.

Houweling, S., Kaminski, T., Dentener, F., Lelieveld, J., and Heimann, M.: Inverse modeling of methane sources and sinks using the adjoint of a global transport model, J. Geophys. Res., 104, 26137–26160, https://doi.org/10.1029/1999JD900428, 1999.

Houweling, S., Bergamaschi, P., Chevallier, F., Heimann, M., Kaminski, T., Krol, M., Michalak, A. M., and Patra, P.: Global inverse modeling of $CH_4$ sources and sinks: An overview of methods, Atmospheric Chem. Phys., 17, 235–256, https://doi.org/10.5194/acp-17-235-2017, 2017.

Lan, X., Basu, S., Schwietzke, S., Bruhwiler, L. M. P., Dlugokencky, E. J., Michel, S. E., Sherwood, O. A., Tans, P. P., Thoning, K., Etiope, G., Zhuang, Q., Liu, L., Oh, Y., Miller, J. B., Pétron, G., Vaughn, B. H., and Crippa, M.: Improved Constraints on Global Methane Emissions and Sinks Using $\delta^{13}C$-$CH_4$, Glob. Biogeochem. Cycles, 35, e2021GB007000, https://doi.org/10.1029/2021GB007000, 2021.

McNorton, J., Wilson, C., Gloor, M., Parker, R. J., Boesch, H., Feng, W., Hossaini, R., and Chipperfield, M. P.: Attribution of recent increases in atmospheric methane through 3-D inverse modelling, Atmospheric Chem. Phys., 18, 18149–18168, https://doi.org/10.5194/acp-18-18149-2018, 2018.

Miller, J. B.: The Carbon Isotopic Composition of Atmospheric Methane and its Constraint on the Global Methane Budget, in: Stable Isotopes and Biosphere - Atmosphere Interactions, edited by: Pataki, D., Ehleringer, J. R., and Flanagan, L. B., Academic Press, 288–310, https://doi.org/10.1016/B978-012088447-6/50016-7, 2004.

Nicely, J. M., Canty, T. P., Manyin, M., Oman, L. D., Salawitch, R. J., Steenrod, S. D., Strahan, S. E., and Strode, S. A.: Changes in Global Tropospheric OH Expected as a Result of Climate Change Over the Last Several Decades, J. Geophys. Res. Atmospheres, 123, 10,774-10,795, https://doi.org/10.1029/2018JD028388, 2018.

Patra, P. K., Houweling, S., Krol, M., Bousquet, P., Belikov, D., Bergmann, D., Bian, H., Cameron-Smith, P., Chipperfield, M. P., Corbin, K., Fortems-Cheiney, A., Fraser, A., Gloor, E., Hess, P., Ito, A., Kawa, S. R., Law, R. M., Loh, Z., Maksyutov, S., Meng, L., Palmer, P. I., Prinn, R. G., Rigby, M., Saito, R., and Wilson, C.: TransCom model simulations of $CH_4$ and related species: linking transport, surface flux and chemical loss with $CH_4$ variability in the troposphere and lower stratosphere, Atmospheric Chem. Phys., 11, 12813–12837, https://doi.org/10.5194/acp-11-12813-2011, 2011.

Rigby, M., Montzka, S. A., Prinn, R. G., White, J. W. C., Young, D., O'Doherty, S., Lunt, M. F., Ganesan, A. L., Manning, A. J., Simmonds, P. G., Salameh, P. K., Harth, C. M., Mühle, J., Weiss, R. F., Fraser, P. J., Steele, L. P., Krummel, P. B., McCulloch, A., and Park, S.: Role of atmospheric oxidation in recent methane growth, Proc. Natl. Acad. Sci., 114, 5373–5377, https://doi.org/10.1073/pnas.1616426114, 2017.

Sanderson, M. G.: Biomass of termites and their emissions of methane and carbon dioxide: A global database, Glob. Biogeochem. Cycles, 10, 543–557, https://doi.org/10.1029/96GB01893, 1996.

Saunois, M., Stavert, A. R., Poulter, B., Bousquet, P., Canadell, J. G., Jackson, R. B., Raymond, P. A., Dlugokencky, E. J., Houweling, S., Patra, P. K., Ciais, P., Arora, V. K., Bastviken, D., Bergamaschi, P., Blake, D. R., Brailsford, G., Bruhwiler, L., Carlson, K. M., Carrol, M., Castaldi, S., Chandra, N., Crevoisier, C., Crill, P. M., Covey, K., Curry, C. L., Etiope, G., Frankenberg, C., Gedney, N., Hegglin, M. I., Höglund-Isaksson, L., Hugelius, G., Ishizawa, M., Ito, A., Janssens-Maenhout, G., Jensen, K. M., Joos, F., Kleinen, T., Krummel, P. B., Langenfelds, R. L., Laruelle, G. G., Liu, L., Machida, T., Maksyutov, S., McDonald, K. C., McNorton, J., Miller, P. A., Melton, J. R., Morino, I., Müller, J., Murguia-Flores, F., Naik, V., Niwa, Y., Noce, S., O'Doherty, S., Parker, R. J., Peng, C., Peng, S., Peters, G. P., Prigent, C., Prinn, R., Ramonet, M., Regnier, P., Riley, W. J., Rosentreter, J. A., Segers, A., Simpson, I. J., Shi, H., Smith, S. J., Steele, L. P., Thornton, B. F., Tian, H., Tohjima, Y., Tubiello, F. N., Tsuruta, A., Viovy, N., Voulgarakis, A., Weber, T. S., van Weele, M., van der Werf, G. R., Weiss, R. F., Worthy, D., Wunch, D., Yin, Y., Yoshida, Y., Zhang, W., Zhang, Z., Zhao, Y., Zheng, B., Zhu, Q., Zhu, Q., and Zhuang, Q.: The Global Methane Budget 2000--2017, Earth Syst. Sci. Data, 12, 1561–1623, https://doi.org/10.5194/essd-12-1561-2020, 2020.

Turner, A. J., Frankenberg, C., Wennberg, P. O., and Jacob, D. J.: Ambiguity in the causes for decadal trends in atmospheric methane and hydroxyl, Proc. Natl. Acad. Sci., 114, 5367–5372, https://doi.org/10.1073/pnas.1616020114, 2017.

Weir, B., Ott, L. E., Collatz, G. J., Kawa, S. R., Poulter, B., Chatterjee, A., Oda, T., and Pawson, S.: Bias-correcting carbon fluxes derived from land-surface satellite data for retrospective and near-real-time

assimilation systems, Atmospheric Chem. Phys., 21, 9609–9628, https://doi.org/10.5194/acp-21-9609-2021, 2021.

Worden, J. R., Bloom, A. A., Pandey, S., Jiang, Z., Worden, H. M., Walker, T. W., Houweling, S., and Röckmann, T.: Reduced biomass burning emissions reconcile conflicting estimates of the post-2006 atmospheric methane budget, Nat. Commun., 8, 2227–2227, https://doi.org/10.1038/s41467-017-02246-0, 2017.

---

## Author Response (AR2)

Dear Editor,

In a previous upload, we detailed our response to the referees' comments as well as explained our responses to you. Please find here our response to your concerns about the GCP budget, which you would like us to revise.

Table 3 of Saunois et al (2020) quotes emissions for three periods, (i) 2000—2009, (ii) 2008—2017, and (iii) 2017. The first period, 2000—2009, is directly comparable to our inversions and we have computed 2000—2009 means from our inversions and reported those in the newly added §4.2. However, our inversions end in 2016. Therefore, we constructed the 2008—2016 means of GCP emissions from periods (ii) and (iii) reported by Saunois et al (2020). These are the GCP numbers quoted in §4.2. Finally, we used the GCP spreadsheet at https://doi.org/10.18160/GCP-CH4-2019 instead of the numbers in Table 3 of Saunois et al (2020) to calculate means, which resulted in small differences due to rounding. For example, the 2000—2009 mean bottom-up fossil fuel emissions computes to 110.51 Tg/yr from the spreadsheet, which is rounded (incorrectly, in our opinion) to 110 Tg/yr in the paper. We round it to 111 Tg/yr when we quote the GCP budget. This possibility of small differences due to rounding is acknowledged in Table 3 of Saunois et al (2020) as well.

We submit that the small inconsistencies and differences the editor has noticed are due to these two factors, namely (i) using the GCP spreadsheet instead of the rounded numbers quoted in the paper, and (ii) calculating and using the 2008—2016 GCP means, which are not reported directly in Saunois et al (2020). We realize that at first glance our GCP numbers and the numbers quoted in Saunois et al (2020) may seem inconsistent. We have therefore added lines 515—520 in §4.2 of the revised manuscript for explanation. We hope this will address the editor's concerns.

Kind regards,

Sourish Basu
(on behalf of all co-authors)

References

Saunois, M., Stavert, A. R., Poulter, B., Bousquet, P., Canadell, J. G., Jackson, R. B., Raymond, P. A., Dlugokencky, E. J., Houweling, S., Patra, P. K., Ciais, P., Arora, V. K., Bastviken, D., Bergamaschi, P., Blake, D. R., Brailsford, G., Bruhwiler, L., Carlson, K. M., Carrol, M., Castaldi, S., Chandra, N., Crevoisier, C., Crill, P. M., Covey, K., Curry, C. L., Etiope, G., Frankenberg, C., Gedney, N., Hegglin, M. I., Höglund-Isaksson, L., Hugelius, G., Ishizawa, M., Ito, A., Janssens-Maenhout, G., Jensen, K. M., Joos, F., Kleinen, T., Krummel, P. B., Langenfelds, R. L., Laruelle, G. G., Liu, L., Machida, T., Maksyutov, S., McDonald, K. C., McNorton, J., Miller, P. A., Melton, J. R., Morino, I., Müller, J., Murguia-Flores, F., Naik, V., Niwa, Y., Noce, S., O'Doherty, S., Parker, R. J., Peng, C., Peng, S., Peters, G. P., Prigent, C., Prinn, R., Ramonet, M., Regnier, P., Riley, W. J., Rosentreter, J. A., Segers, A., Simpson, I. J., Shi, H., Smith, S. J., Steele, L. P., Thornton, B. F., Tian, H., Tohjima, Y., Tubiello, F. N., Tsuruta, A., Viovy, N., Voulgarakis, A., Weber, T. S., van Weele, M., van der Werf, G. R., Weiss, R. F., Worthy, D., Wunch, D., Yin, Y., Yoshida, Y., Zhang, W., Zhang, Z., Zhao, Y., Zheng, B., Zhu, Q., Zhu, Q., and Zhuang, Q.: The Global Methane Budget 2000--2017, Earth Syst. Sci. Data, 12, 1561–1623, https://doi.org/10.5194/essd-12-1561-2020, 2020.

---

## Author Response (AR3)

Dear Editor,

Thank you for suggesting the corrections. We have now added definitions of $\delta_s$ and $\delta_a$ when they are first introduced in §2.1.

Kind regards,

Sourish Basu
(on behalf of all co-authors)